# Intercellular communication atlas reveals Oprm1 as a neuroprotective factor for retinal ganglion cells

Cheng Qian [1,7], Ying Xin [2,7], Cheng Qi [1], Hui Wang [3], Bryan C. Dong[4], Donald J. Zack[2], Seth Blackshaw [2,5], Samer Hattar [3], Feng-Quan Zhou [1,5,6] ✉ & Jiang Qian [2] ✉

Previous studies of neuronal survival have primarily focused on identifying intrinsic mechanisms controlling the process. This study explored how intercellular communication contributes to retinal ganglion cell (RGC) survival following optic nerve crush based on single-cell RNA-seq analysis. We observed transcriptomic changes in retinal cells in response to the injury, with astrocytes and Müller glia having the most interactions with RGCs. By comparing RGC subclasses characterized by distinct resilience to cell death, we found that the high-survival RGCs tend to have more ligand-receptor interactions with neighboring cells. We identified 47 interactions stronger in high-survival RGCs, likely mediating neuroprotective effects. We validated one identified target, the μ-opioid receptor (Oprm1), to be neuroprotective in three retinal injury models. Although the endogenous Oprm1 is preferentially expressed in intrinsically photosensitive RGCs, its neuroprotective effect can be transferred to other subclasses by pan-RGC overexpression of Oprm1. Lastly, manipulating the Oprm1 activity improved visual functions in mice.

Neuronal cell death can lead to irreversible loss of sensory, motor, and cognitive functions[1,2]. Enhancing neuroprotection is, therefore, one of the major strategies for potentially delaying the development of neurological diseases. While previous studies have identified many genes and signaling pathways governing neuronal cell death or neuroprotection[3–6], most attention focused on intrinsic factors and cell-autonomous regulatory mechanisms. However, tissues contain diverse types of cells forming and regulating their microenvironment collectively[7,8]. Although the importance of the microenvironment has been well recognized and extensively investigated in the context of stem cell biology, immunology, and cancer[9–11], it is much less clear how it functions in the mature nervous system to regulate diverse neurological functions.

One primary form of intercellular communication in the tissue microenvironment is ligand–receptor interactions, potentially including autocrine, paracrine, and endocrine signaling. In the retina, there are many types of cells, such as retinal ganglion cells (RGCs), amacrine cells, bipolar cells, Müller glia, astrocytes, microglia, photoreceptors, and epithelial cells[12]. Cell–cell interactions in the retina are critical for retinal circuit formation during development and maintaining normal retinal functions in adult animals, but also regulate tissue repair after injury. For instance, amacrine cells can regulate RGC maturation and their intrinsic axon growth ability during retinal development via direct cell–cell contact[13]. In adult mammals, microglia have been shown to limit the spontaneous neurogenic ability of Müller glia via secretion of TNF-α[14]. However, these studies mainly focus on

[1]Department of Orthopaedic Surgery, Johns Hopkins University School of Medicine, Baltimore, MD, USA. [2]Department of Ophthalmology, Johns Hopkins University School of Medicine, Baltimore, MD, USA. [3]Section on Light and Circadian Rhythms, National Institute of Mental Health, Bethesda, MD, USA. [4]Neuroscience Study Program, Krieger School of Arts & Sciences, Johns Hopkins University, Baltimore, MD, USA. [5]Solomon H. Snyder Department of Neuroscience, Johns Hopkins University School of Medicine, Baltimore, MD, USA. [6]Present address: Sir Run Run Shaw Hospital, Zhejiang University School of Medicine, Hangzhou, China. [7]These authors contributed equally: Cheng Qian, Ying Xin. ✉e-mail: fzhou4@zju.edu.cn; jiang.qian@jhmi.edu

interactions between two cell types via a limited number of interactions. Recent advances in single-cell sequencing technologies provide the opportunity to systematically map cell–cell interactions based on the expression of ligands, receptors, and other associated genes in each cell type[15,16].

In this study, we employed the optic nerve crush (ONC) model, widely used to investigate neuronal cell death and survival in the central nervous system[3–5]. This model is particularly suitable for studying intercellular communication since the physical crush injury specifically affects the axons of RGCs, and the alterations in other retinal cells are likely to be triggered through cell–cell communications with the RGCs. To comprehensively explore the cellular effects of ONC, we conducted the single-cell sequencing of all retinal cells at various time points after the procedure. By doing so, we constructed an atlas of cell–cell communication that systematically documents the identities of individual ligand–receptor interactions and their dynamic regulation in response to ONC. Moreover, through a comparative analysis of interactions among different RGC subclasses, each exhibiting varying levels of resilience to cell death, we successfully identified numerous potential neuroprotective ligand–receptor interactions between the RGCs and their neighboring cells.

We identified the μ-opioid receptor, encoded by the *Oprm1* gene, as a neuroprotective factor for RGCs. The μ-opioid receptor is the first discovered opioid receptor and plays a central role in regulating pain, reward, and addictive behaviors[17–19]. We functionally validated Oprm1 as a neuroprotective factor for RGCs by demonstrating that overexpression of Oprm1 in RGCs not only led to significantly increased survival rate of RGCs following several different types of retinal injury but also significantly improved visually-guided perception behavior. Our study establishes an effective strategy to identify functionally important cell–cell interactions in a complex tissue microenvironment.

## Results

### Effect of optic nerve crush on retinal cell transcriptomes
We performed droplet-based single-cell RNA sequencing (scRNA-seq) on cells dissociated from the mouse whole retina collected under sham conditions and at three-time points after the optic nerve crush (ONC) (Fig. 1a). In total, we profiled 56,531 retinal cells, which were clustered and annotated as 14 major cell types based on the known retinal cell type marker genes (Fig. 1b–d. see cell numbers in Supplementary Table 1). For example, *Abca8a, Rpe65, and Opn1mw* are representative marker genes for Müller glia, retinal pigment epithelial cells, and cone cells, respectively (Fig. 1c). The specific expression patterns of the known marker genes demonstrated the high quality of the cell annotation (Fig. 1d).

We first asked if the ONC affected the gene expression in non-RGC retinal cells. To address this question, we compared the gene expression patterns in each retinal cell type before and after the ONC. The results showed that all identified retinal cell types had differential expressed genes (DEGs) in response to ONC with distinct patterns at different time points (Fig. 1e and Supplementary Fig. 1). For instance, in the Müller glia cells (MG), ONC led to down and up-regulation of multiple sets of genes with different time courses (Fig. 1e). We then performed the Gene Ontology (GO) analysis of the genes with altered expression levels in retinal cell types (Fig. 1f and Supplementary Fig. 1). Interestingly, genes associated with chromatin remodeling transiently downregulated in MG (Fig. 1f), in line with their inability to reprogram for spontaneous regeneration after retinal injuries[20]. In addition, the genes associated with cytokine production and immune response were enriched in microglia (Supplementary Fig. 1b), consistent with their roles as resident immune cells in retinas. Moreover, the GO analyses have demonstrated a temporal increase in glycolytic activity, ribosomal subunit assembly, cytoplasmic translation, peptide biosynthesis, and oxidative phosphorylation processes within the two-day

timeframe following ONC. This increase was observed in retinal ganglion cell (RGC) neighboring cell types, including Müller glial cells (MG), astrocytes, microglia, or amacrine cells (Fig. 1f and Supplementary Fig. 1a–c). These findings are associated with observations that cellular stresses may induce a metabolic shift toward glycolysis to facilitate a transient boost in macromolecule biosynthesis[21,22]. The inducement for such changes in gene transcription within those non-RGC cell types is likely through ligands expressed from RGCs in response to the ONC. Indeed, genes encoding ligands and receptors, as a group, are more likely to be differentially expressed than genes of all functional categories (Fig. 1g and Supplementary Fig. 1).

### Ligand–receptor interactions induced by ONC
To identify individual ligand–receptor interactions between RGCs and other retinal cell types, we integrated our scRNA-seq dataset with a publicly available dataset for purified RGCs, obtained at the same time points post-ONC (12, 24, and 48 h)[4]. The mean gene expression profile of the 1055 RGC cells from our collection and that of 86,426 RGCs obtained by Tran et al.[4] exhibited a high degree of correlation (Supplementary Fig. 2a). Furthermore, the ligand-receptor interaction strengths were also highly correlated between the two datasets (Supplementary Fig. 2a right panel), indicating that both datasets are compatible. Also, the ONC microsurgeries established in different laboratories were comparable. For subsequent analysis of cell–cell communication between RGCs and retinal cells, we used the scRNA-seq dataset from Tran et al. for RGCs, which encompasses the majority of known RGC subclasses.

We next performed LRLoop, our recently developed computational method for analyzing ligand–receptor-based cell–cell communication, to explore intercellular interactions between RGCs and every other retinal cell type[23]. This method utilized the transcriptome data of both ligand sender and receiver cells, adjusting interaction scores based on the presence of ligand-receptor feedback loops between the cells. LRLoop was first used to identify initial signaling interactions from RGCs to other retinal cells, termed the distress interactions.

Several vital characteristics were associated with these distress interactions. First, while all cell types were receivers of distress signals, based on the number of ligand–receptor pairs, astrocytes, Müller glia, GABAergic amacrine, and vasculature pericytes revealed the most robust interactions (Fig. 2a). The overall interaction strength remains relatively consistent across the four timepoints and is not correlated with the cell counts in the retinal cell types (Supplementary Fig. 2b and c). Interestingly, the interaction strength primarily correlates with their physical proximity to the ganglion cell layer (GCL). For instance, astrocytes and Müller glia, in close contact with the GCL[12], received the most ligand–receptor interactions from RGCs. Second, most cell types utilized common types of ligand–receptor interactions. Specifically, ligands produced by RGCs have their corresponding receptors expressed in most non-RGC retinal cells (Fig. 2a). However, a few distress signals were specific to one cell type (Fig. 2b), in which microglia theoretically received unique interactions in line with their role as the resident inflammatory cells[24] (Fig. 2a and b). Third, the strengths of distress ligand–receptor interactions, projected by the interaction scores, displayed transient as well as prolonged dynamics after ONC (Fig. 2c). As an example, interactions from RGCs to astrocytes could be grouped into four primary categories based on the similarity of the dynamic interaction patterns (Fig. 2c). The first two groups responded transiently to the injury, being repressed or activated, but mostly returning to their original states one day after ONC. Another two groups displayed quick but sustained or slowly activated interactions (Fig. 2c). Finally, the biological processes associated with these four groups showed considerable overlaps, including axonogenesis, neuron projection guidance, and axon guidance (Fig. 2d).

We subsequently identified the reciprocal, responsive ligand–receptor interactions sent from the other retinal cells back to

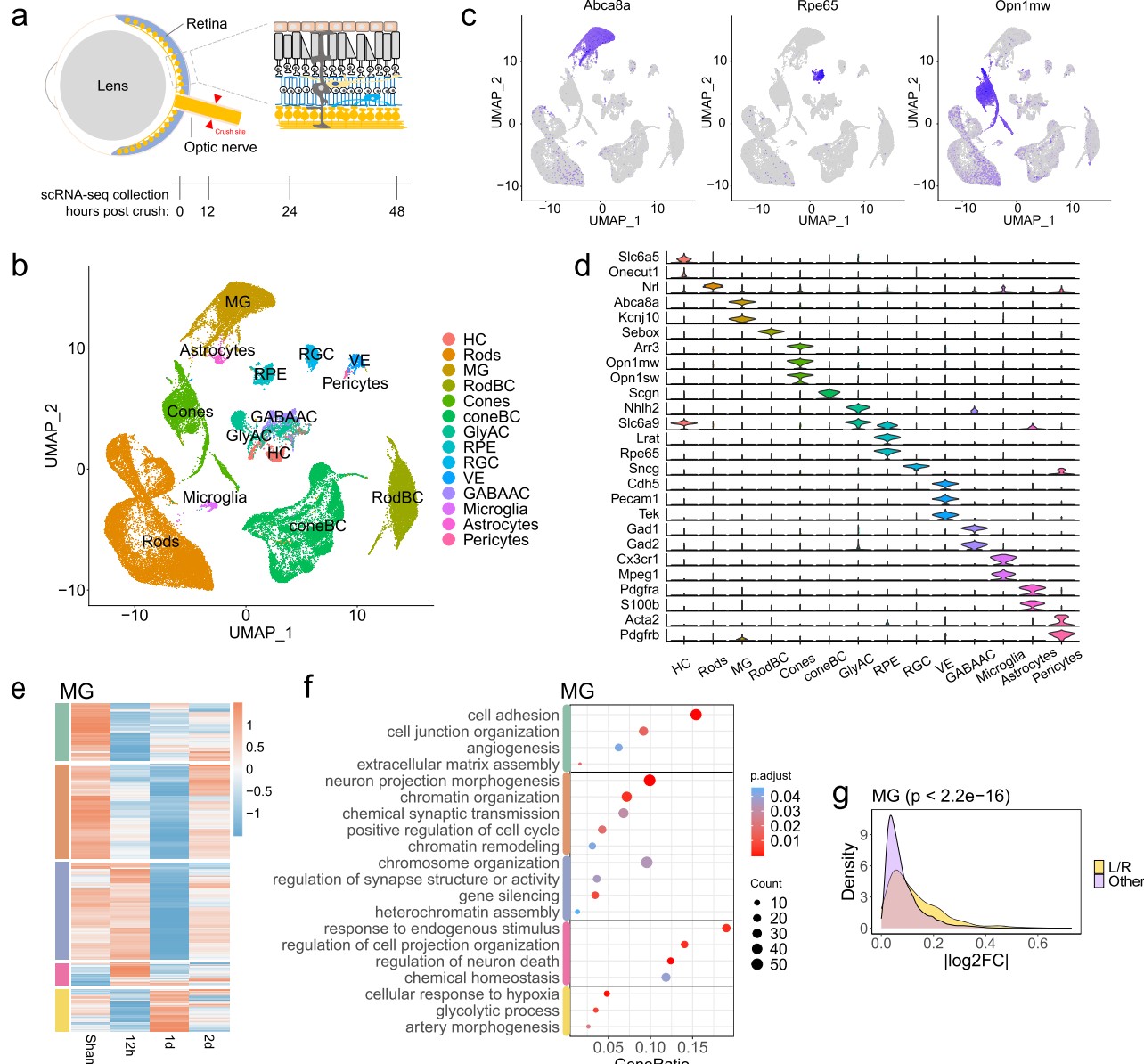

**Fig. 1 | Retinal cells in response to optic nerve crush. a** Schematic diagram illustrates the crush site on the optic nerve and the timeline for retinal tissue collection and dissociation. **b** UMAP exhibits retinal cell types identified with scRNA-seq data of whole retinal tissues of mice. A total of 56,531 retinal cells were profiled to annotate 14 major cell types. **c** UMAP shows the expression pattern of *Abca8a, Rpe65*, or *Opn1mw* in whole mouse retinal cells, which represent the marker genes for Müller glia (MG), retinal pigment epithelium (RPE) cells, and cone cells, respectively. **d** Expression patterns of representative known marker genes in retinal cell types. **e** Heatmap exhibits differentially expressed genes (DEGs) in Müller glia (MG), for example, at different time points before and after ONC. **f** Gene

ontology analysis (GO) reveals the representative biological processes based on DEG patterns shown in panel (**e**). The *p*-values are based on a hypergeometric test, adjusted by the Benjamini–Hochberg method. **g** Density plot illustrates the absolute fold-change (log2FC) of genes expressed in MG (detection rate > 0.1). The highest absolute |log2FC| value among three comparisons (12 h vs. control, 24 h vs. control, and 48 h vs. control) was calculated for each gene. Ligand and receptor genes (L/R) are grouped in yellow, while the other genes are shown in light purple for comparison. The *p*-values are based on the Kolmogorov–Smirnov test (two-sided).

RGCs. Based on our bioinformatics analyses, some of these responsive interactions might be triggered by the distress interactions, thereby forming a cell–cell signaling loop (Fig. 2e and Supplementary Fig. 2d). For instance, VEGF-A is known to be neuroprotective for RGCs under stress[25,26]. Our analysis showed that the secretion of VEGF-A from astrocytes could potentially be initiated by the Bdnf-Ntrk2 distress interaction from RGCs to astrocytes, as these interactions formed a loop through signaling and regulatory network (Fig. 2e, more examples in Supplementary Fig. 2d). We discovered that the overall scores of the responsive interactions among the retinal cell types correlated with the distress interactions sent from RGCs (Supplementary Fig. 2e

and f). Dynamics patterns similar to distress interactions were also noted for the responsive interactions (Supplementary Fig. 2g). These interactions also play functional roles in axonogenesis, neuron projection guidance, and synapse organization (Supplementary Fig. 2h). Together, we established a retinal cell–cell communication atlas before and after a stressed condition of ONC with spatiotemporal information.

## Protective interactions associated with RGC survival
Prior studies have shown differential resilience to cell death following stress and damage in distinct transcriptome- and function-related RGC

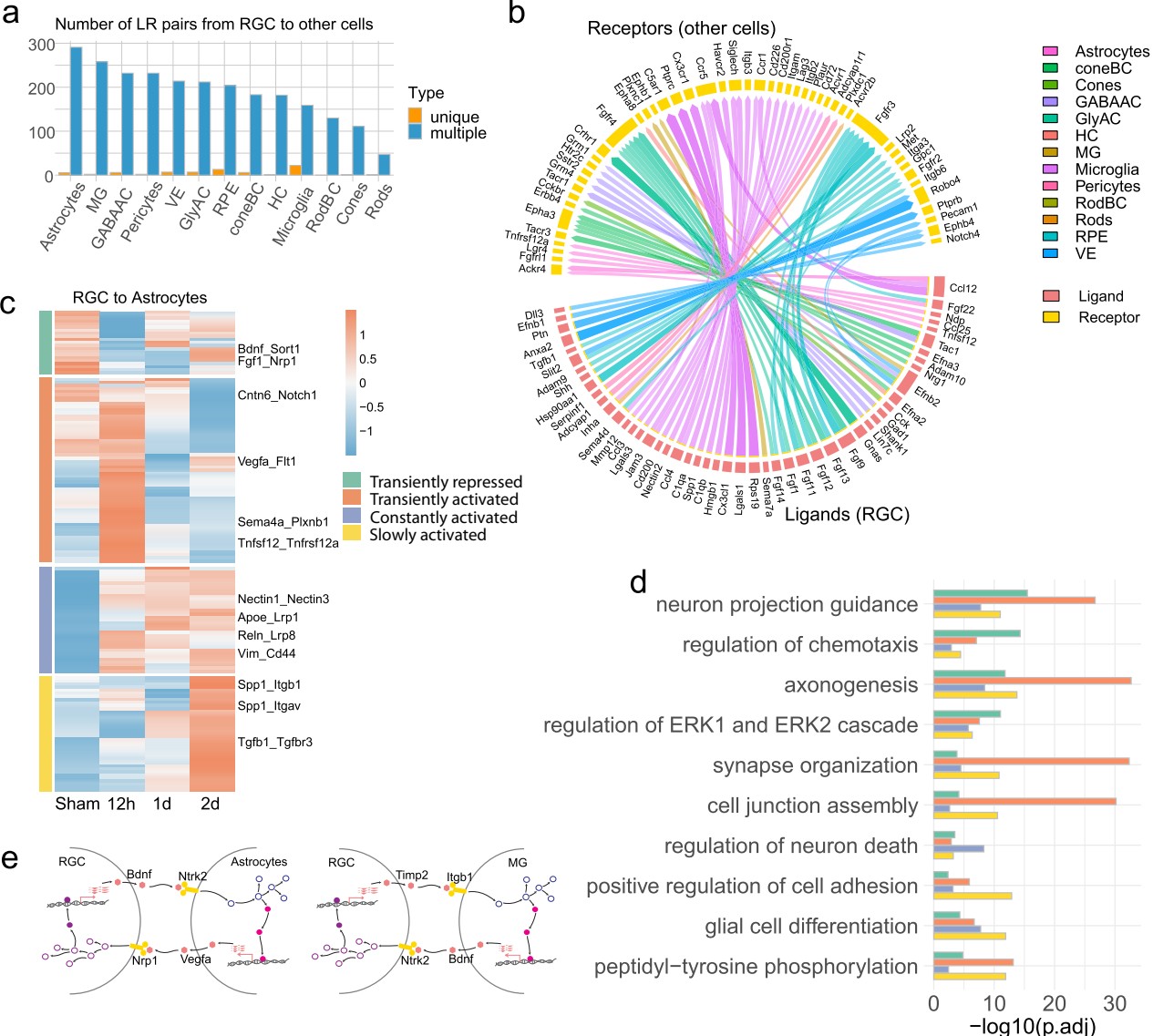

**Fig. 2 | Distress interactions from RGCs to other retinal cells. a** The shared and unique ligand–receptor interactions summed across four-time points identified from RGCs to other retinal cell types. Interactions identified in multiple and unique cell types are dark blue and orange. **b** The specific ligand–receptor interactions identified from RGCs to a unique retinal cell type, which are shown in orange color columns in panel (**a**). Genes in the bottom half of the circle are the ligands from RGCs, and the genes in the top half of the circle are the receptors in other retinal cells. The colors of the connecting edges represent the receiver cell types. **c** Heatmap shows ONC-induced dynamical patterns of variable interactions sent from RGCs to astrocytes, as one example, across time points (the fold change

(FC) > 1.2, of the interaction scores between any two-time points). Four major groups were identified based on their dynamic patterns. **d** Gene Ontology analysis (GO) reveals biological processes of variable ligand-receptor interactions in each group, as shown in panel (**c**) GO enrichment was calculated with all expressed genes (detection rate > 0.1) in either RGCs or astrocytes at any time point as the background. The *p*-values are based on a hypergeometric test, adjusted by the Benjamini–Hochberg method. **e** Schematic examples of the cell–cell interaction feedback loops. A ligand from the sender cell interacts with a receptor on the receiver cell, triggering gene transcription in receiver cells, in which some ligand genes are transcribed and sent back to the sender cells.

subclasses[4,27–29]. Three major RGC subclasses (i.e., ipRGC, αRGC, and Gpr88RGC) demonstrated the highest survival rates post-ONC, while other RGC subclasses were susceptible to injury, resulting in cell death[4,23]. We sought to determine if intercellular communications between different RGC subclasses and retinal cells contribute to the distinct survival rates of RGC subtypes. By calculating the score of ligand–receptor interactions ($S_{LR}$), established in our recent methodology study[23], we discovered that the overall interaction scores (sum of $S_{LR}$) were higher for three high-survival RGC subclasses compared to low-survival RGCs (Fig. 3a). This finding suggested that in the tissue microenvironment, neighboring retinal cells participate in facilitating RGC survival in addition to previously identified cell-autonomous mechanisms[3–6]. Interestingly, the high-survival RGC subclasses

expressed more significant numbers of ligand and receptor genes (Supplementary Fig. 3a), suggesting high-survival RGC subclasses might be intrinsically programmed to foster more communication with their surroundings.

Next, we tried to determine the identities of individual ligand-receptor interactions sent from retinal cell types to RGCs that might contribute to the different survival rates of RGC subclasses. To do so, we calculated the $S_{LR}$ of ligand–receptor interactions received by various RGC subclasses. By comparing the difference in interaction strength (i.e., mean $S_{LR}$ values) between high- and low-survival subclasses, we discovered several interactions were stronger (i.e., larger $S_{LR}$) in high-survival RGCs than in low-survival RGCs, which are likely to be neuroprotective interactions (Fig. 3b). Interestingly, we did not

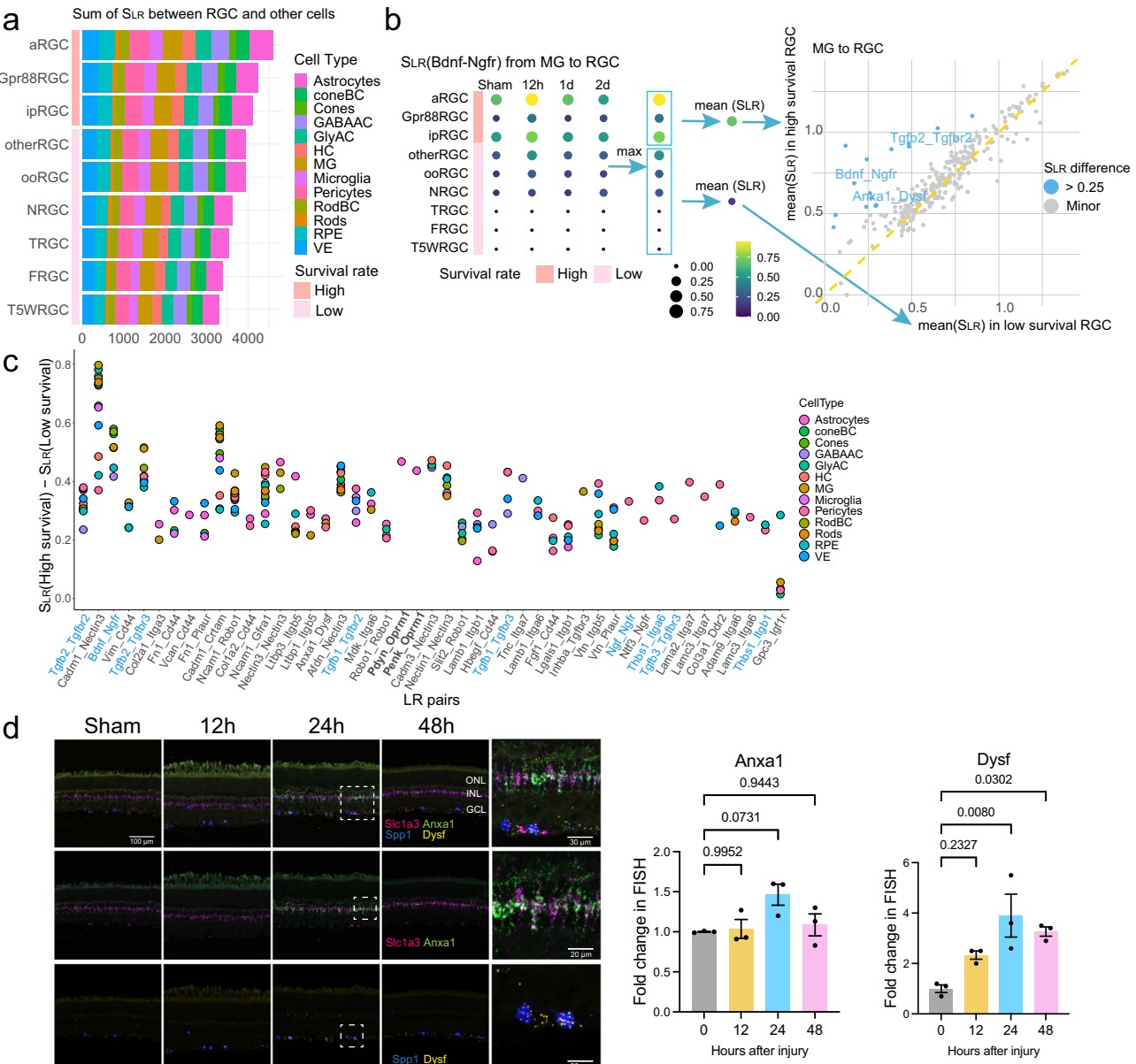

**Fig. 3 | Prediction of neuroprotective interactions for RGCs. a** The sum of interaction scores ($S_{LR}$) between each RGC subclass and other retinal cell types (from and to RGCs). **b** The workflow for calculating protective interactions sent to RGCs by Müller glia (MG), as one example cell type. Left: using the interaction Bdnf-Ngfr as an example, the interaction scores ($S_{LR}$) were calculated between each RGC subclass and MG at four-time points. The maximal scores across four-time points (each row) were obtained to calculate the mean values. The mean values of interaction scores were then computed separately for high- and low-survival RGC subclasses (the max $S_{LR}$ scores enclosed in two rectangles). Right: For each ligand–receptor pair, the two calculated mean values, one (*y*-axis) for the high- and the other (*x*-axis) for the low-survival RGC subclasses, were plotted for comparison. Each dot represents a ligand-receptor interaction. Blue dots are the interactions that are stronger in high–than in low-survival RGC subclasses. Gray dots are the interactions with similar interaction scores between the two categories. **c** Summary of the potentially neuroprotective interactions sent from different retinal cells to RGCs. Similar to the calculation of the MG-to-RGC protective interactions illustrated in panel (**b**). The top 47 pairs of protective interactions were identified based on the $S_{LR}$ score difference between high- and low-survival RGCs. Some well-known factors with neuroprotective effects are colored blue, and Oprm1 receptor-related interactions are labeled with bold font. **d** Validation of the Anxa1–Dysf interaction, from MG to αRGC, using in situ hybridization. The last column of micrographs are high-magnification images of those boxed areas in the 24 h column. Scale bars are labeled in specific micrographs. Quantification of the fluorescent intensities of Anxa1 and Dysf are presented as mean values ± SEM, *n* = 3 mice, Ordinary one-way ANOVA, Tukey's multiple comparisons test, with a single pooled variance. Source data are provided as a Source Data file.

identify neurotoxic interactions (i.e., those with particularly more significant $S_{LR}$ scores in low survival RGC subclasses). For example, for the signals from Müller glia to RGCs, we identified 15 interactions (shown in blue dots in Fig. 3b right panel) favored in high-survival subclasses but none in the low-survival subclasses. Among these 15 ligand–receptor pairs, the roles of Tgfb2-Tgfbr2 and Bdnf-Ngfr signaling in RGC survival have been previously reported, indicating that our analysis was successful in identifying some known neuroprotective

interactions[30–32]. Similar results were also observed between additional retinal cell types and RGCs (Supplementary Fig. 3b).

Combining the neuroprotective interactions from all retinal cell types analyzed, we identified 47 non-redundant interactions that potentially promote RGC survival (Fig. 3c and Supplementary Table 2). Most of these neuroprotective interactions are initiated by ligands expressed in multiple retinal cell types (Fig. 3c). For instance, the cell adhesion molecule Cadm1, which is expressed on almost all retinal cell

types included in the study, is a ligand for the receptor Nectin3, which is preferentially expressed in the αRGCs. Conversely, some neuroprotective interactions were explicitly sent from specific types of retinal cells. For instance, the Penk-Oprm1, Tnc-Ltga7, and Lama2-Itga7 were protective interactions sent to high-survival RGCs from a specific sender cell type of astrocytes, GABAergic amacrine cells or pericytes, respectively. Our predicted protective interactions include numerous known pro-survival factors, including TGF-β, BDNF, NGF, and thrombospondin, etc.[27,30–33].

We validated the expression pattern of Anxa1-Dysf, one interaction with dynamic changes following ONC, primarily between Müller glia (MG) and αRGCs, using the fluorescence in situ hybridization (Fig. 3d and Supplementary Fig. 3c). The ligand Annexin A1 (Anxa1) is a known regulator in inflammation[34] and can undergo either externalized or secreted, functioning in an autocrine, paracrine and juxtacrine manner[35]. Its mRNA was detected in the MG, co-localizing with MG marker Slc1a3. The receptor Dysferlin (Dysf), which is involved in muscle regeneration and accumulation in brains of Alzheimer's disease[36], was detected in αRGCs, co-localizing with Spp1 (Osteopontin)[28,37]. One day post-ONC, a 3.9- and 1.4-fold increase of *Dysf* and *Anxa1* mRNA expression were observed at the transcriptional level compared to the sham condition, respectively (Fig. 3d). These results are consistent with the scRNA-seq data, which indicated a 2.6-fold increase for *Dysf* in αRGCs and a 1.1-fold increase for *Anxa1* in Müller glia following injury.

### Features of the protective interactions

To gain further insights into how cell–cell communications regulate RGC survival, we examined the expression patterns of ligands or receptors from these neuroprotective interactions in different RGC subclasses. We discovered that the three high-survival subclasses each had their distinct sets of receptor and ligand gene expression (Fig. 4a and Supplementary Fig. 4a), suggesting that these RGCs might be intrinsically equipped to use subclass-specific ligand–receptor pairs for intercellular communications with the tissue microenvironment. When we examined a group of neuroprotective interactions identified with astrocytes as sender cells, we found that while all three high-survival RGC subclasses shared some ligand-receptor pairs, the others were preferentially enriched in each RGC subclass (Fig. 4b). For instance, the Fn1–Plaur interaction was the strongest in αRGC, while the Fgf13–Scn5a pair showed a preference in Gpr88RGC. Our analysis suggests that the high-survival RGCs might use distinct cell–cell communications to promote resistance to cell death.

We next tested whether these protective interactions were preset in uninjured high-survival RGCs or only induced by the ONC injury. The results showed some protective signaling interactions were preferentially active in high-survival RGCs before ONC injury. For instance, Tgfb2-Tgfbr2 from astrocytes to RGCs was significantly more substantial in high-survival αRGCs in uninjured mice and remained essentially unchanged post-injury (Fig. 4c). Some protective interactions, however, were upregulated following ONC. For example, Fn1-Cd44 signaling from astrocytes to αRGCs was upregulated post-ONC injury, though it was already stronger in uninjured high-survival RGCs. In contrast, these interactions were weak both before and after ONC in the low survival RGCs (i.e., in the T-RGCs, pink lines in Fig. 4c). The neuroprotective interactions can be therefore classified into two categories based on their dynamic patterns: intrinsic and induced (i.e., from astrocytes to RGCs, Fig. 4d). Similar features of the neuroprotective ligand–receptor interactions were observed in additional non-RGC retinal cells-to-RGC interactions, such as those from GABAergic amacrine cells (GABA-AC) to RGCs and from Müller glia (MG) to RGCs (Supplementary Fig. 4b).

Finally, we asked whether these protective interactions were primarily achieved through paracrine or autocrine signaling. To address this question, we analyzed the expression patterns of 38 ligands

involved in the 47 protective interactions (Fig. 4e). Many ligands were predominantly expressed in the non-RGC retinal cells, also because the non-RGC retinal cells outnumbered RGCs in retinas, which suggested that the neuroprotective interactions preferentially function through the paracrine signaling. Some ligands (i.e., Fgf1 and Ncam1) were expressed in RGCs and non-RGC retinal cells, suggesting that the associated interactions could be autocrine or paracrine.

### Oprm1 promotes RGC survival

From those top 47 ligand-receptor interactions that may contribute to RGC survival, we selected interactions involving Oprm1 for functional validation. The *Oprm1* gene encodes the μ-Opioid receptor. Our data (Fig. 3c, Supplementary Table 2, and sequencing dataset deposited and visualized at the Broad Institute Single Cell portal with accession number SCP2423) and other public datasets[16] showed the endogenous ligands for Oprm1 including the proenkephalin (Penk), prodynorphin (Pdyn), or proopiomelanocortin (Pomc) were expressed by multiple ocular cell types, including astrocytes, Müller glial cells, vascular endothelial and eyecup ciliary cells.

We first examined if ectopic overexpression of *Oprm1* selectively in pan-RGCs could impact RGC survival using two established in vivo cell death models, ONC and the N-Methyl-D-aspartic acid (NMDA) excitotoxicity models[3,20,38]. vGlut2-Cre;LSL-Sun1GFP mice were infected with AAV2-FLEX-Sun1GFP (Sham) or AAV2-FLEX-hOprm1-mCherry to achieve pan-RGC specific overexpression of Oprm1. The ONC was performed two weeks after the viral infection. The cell survival rate was assessed 5 or 14 days post-ONC (Fig. 5a). We selected these time points based on the observation that approximately 50% of RGCs survived at day five post-ONC. In contrast, 15–20% of RGCs remained viable 14 days after ONC injury[3,4]. These time points enabled us to investigate the neuroprotective effect of Oprm1 at either early or late stages of ONC. Oprm1 overexpression increased cell survival rate from 63.8% to 85.4%, and from 18.7% to 36.0%, at 5 and 14 days post-ONC, respectively (Fig. 5b–d, Supplementary Fig. 5a). Given the quick cell death of RGCs induced by the NMDA treatment, we selected seven days post-NMDA treatment to elucidate the effect of Oprm1 on late-stage NMDA toxicity (Fig. 5e). Oprm1 overexpression increased cell survival from 10.7% to 23.6% in the 7-day NMDA toxicity model (Fig. 5f and g, and Supplementary Fig. 5b).

We next used pharmacological reagents to examine the neuroprotective effect of Oprm1 activation. The impact of Oprm1 agonist morphine on RGC protection was evaluated by daily intraperitoneal injection before and after ONC injury (Fig. 5h). The morphine treatment significantly increased the RGC survival rate from 57.2% to 71.3% at 5 days post-ONC (Fig. 5i and j), which might partially explain a previous study indicating that morphine can preserve RGC function in a rat model[39]. Conversely, we utilized Naloxone, a potent Oprm1 antagonist, to test if the Oprm1 activation was necessary for RGC survival. Since Oprm1 is preferentially expressed in the high-survival ipRGC subclass, we focused on determining how Naloxone affected ipRGC survival following NMDA treatment. The survival rate of ipRGCs decreased from 77.8% to 62.4% following intraperitoneal naloxone injection (Fig. 5k–m, Supplementary Fig. 5c). Similar reduction in the survival rate of ipRGCs was also observed in the ONC model (Supplementary Fig. 5d–f). Collectively, overexpression of Oprm1 showed significant neuroprotective effects on RGCs in two models of retinal injuries.

### Universal protective effect of Oprm1 on different RGC subclasses

To gain insights into molecular mechanisms by which Oprm1 protected RGCs, we performed the single-nucleus RNA sequencing (snRNA-seq) on sorted pan-RGC nuclei from retinas under ONC operation. We collected tissue samples from three conditions: uninjured, five days following ONC, and 5 days following ONC with Oprm1

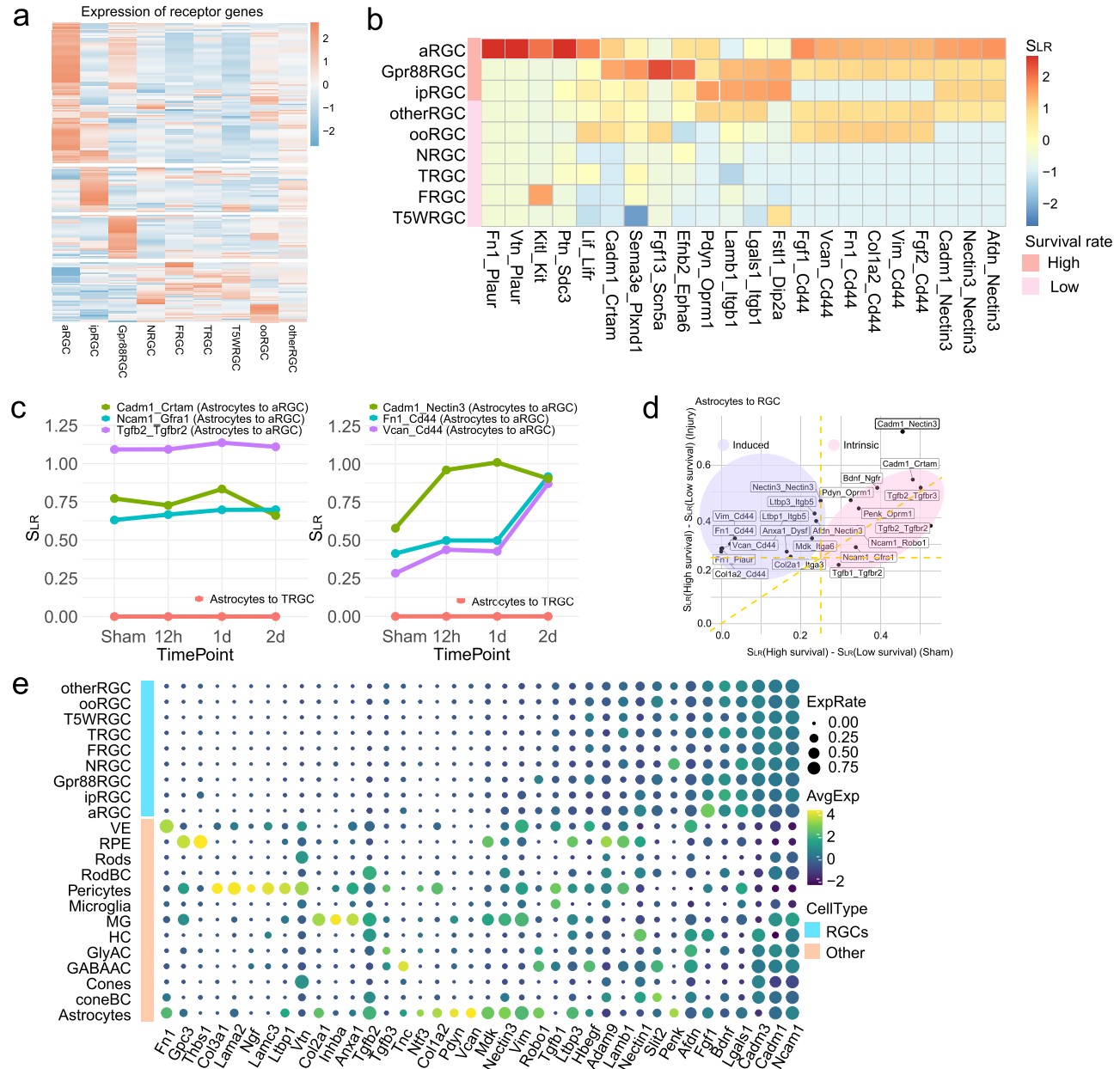

**Fig. 4 | Features of the protective interactions. a** Heatmap shows the average expression level of the genes of various receptors in different RGC subclasses. **b** The top 20 protective ligand-receptor interaction pairs identified in the astrocyte-to-RGC interactions are one example. The heatmap scale represents the interaction score $S_{LR}$. **c** Examples of three preset (left) and three induced (right) protective interactions are from astrocytes to αRGCs. Preset protective interactions are characterized by constantly higher $S_{LR}$ in the high-survival RGC subclasses than in the low-survival subclasses. In contrast, the inducible type of protective interactions becomes stronger after injury. Shown in pink lines, these interactions are almost undetectable from astrocytes to T-RGC, a low-survival RGC subclass. **d** Summary of the preset and induced protective interactions. The *X*-axis is the difference between interaction scores in high- and low-survival subclasses before injury, and the *Y*-axis is the difference between interaction scores in high- and low-survival subclasses after injury. **e** Autocrine vs. paracrine. For the ligand genes involved in protective interactions with receptors expressed in high-survival RGCs, the dot plot summarizes expression levels of ligand genes by RGC subclasses (top) and other retinal cell types (bottom) at 12 h post-ONC.

overexpression. The enrichment to obtain pan-RGC nuclei was successful (Supplementary Fig. 6a and b), and a total of 6348, 3314, and 6141 RGC nuclei were profiled under each condition, respectively. All major RGC subclasses were detected (Fig. 6a) based on the expression profiles of subclass marker genes. In line with our cell–cell communication atlas, we observed that the endogenous Oprm1 was predominantly expressed in the ipRGCs and a subset of αRGCs (Fig. 6a and b). The immunohistochemistry staining against five major RGC subclass makers in the whole mount retina samples of the Oprm1-mCherry reporter mice revealed that Oprm1 was predominantly expressed in Opn4-positive RGCs (86.83% coverage) and 46.58% coverage in Spp1-positive RGCs (Supplementary Fig. 7a and b). Conversely, Oprm1 was rarely expressed in the Tbr1, Satb1, or Foxp2-positive RGCs (1.46%, 1.43%, and 1.61% coverage, respectively) (Supplementary Fig. 7a and b). Since the Oprm1-mCherry reporter mouse strain has the mCherry fused to the C-terminal of Oprm1 receptor (Supplementary Fig. 7c), the mCherry tag reflected the localization of endogenous Oprm1 in the somas and axons of Opn4-positive and Spp1-positive RGCs (Supplementary Fig. 7a). Specifically, among the Oprm1 expressing RGCs, 40.28% cells were Opn4 and Spp1 double-

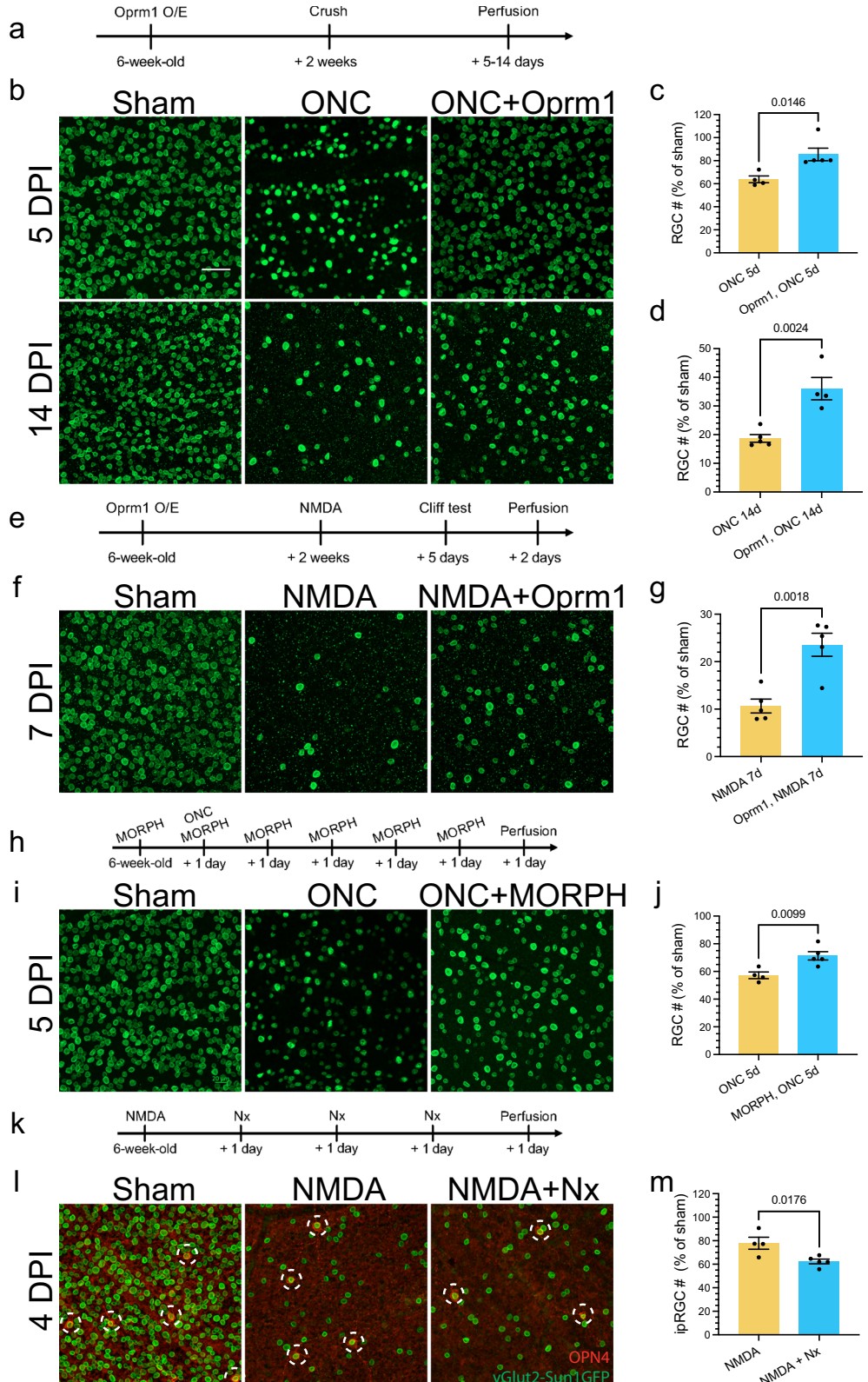

positive, 29.74% RGCs were Opn4+Spp−, 10.10% RGCs were Opn4−Spp1+ and 19.88% RGCs were Opn4 and Spp1 double-negative (Fig. 6c and d). In contrast, the virus-transduced ectopic over-expression of human Oprm1, according to the detection of WPRE element in viral vectors, was shown to distribute to all subclasses of RGC, based on either the snRNA-seq assessment (Fig. 6b) or the immunohistochemistry assay (Supplementary Fig. 8a–c).

We then investigated which RGC subclasses were preserved by the ectopic Oprm1 overexpression from the ONC-induced neuronal cell death. We first computed the cell percentages in each RGC subclass to normalize cell counts across each condition. Then we adjusted these percentages in the control ONC and Oprm1-overexpressed ONC samples by 63.8% and 85.4%, respectively, based on total survival rates determined by immunostaining (Fig. 5c). Because most cells within the

**Fig. 5 | Neuroprotective effect of Oprm1 on RGC survival. a** Experimental timeline of AAV2 infection, injury induction, and tissue collection in the ONC model. **b** Neuroprotective effect of Oprm1 overexpression on RGC survival in the ONC model. Pan-RGC marker vGlut2-Sun1GFP reflected RGC numbers under different conditions. DPI, days post-injury. **c** and **d** Quantification of RGC survival rate in ONC model at 5 or 14 days post-ONC. $n = 4$ or 5 mice for independent experimental conditions. **e** Experimental timeline of AAV2 infection, NMDA toxicity induction and tissue collection in the NMDA toxicity model. **f** Neuroprotective effect of Oprm1 overexpression on RGC survival under NMDA toxicity. **g** Quantification of RGC survival rate seven days in NMDA toxicity model. $n = 5$ mice for independent experimental conditions. **h** Experimental timeline of morphine treatment, crush injury induction, and tissue collection. **i** Neuroprotective effect of morphine treatment on RGC survival at five days post-ONC. **j** Quantification of RGC survival

rate. $n = 4$ or 5 mice for independent experimental conditions. **k** Experimental timeline of NMDA toxicity induction, naloxone treatment, and tissue collection. **l** The effect of Oprm1 antagonist Naloxone on ipRGC survival rate in the NMDA toxicity model. Fluorescent micrographs of ipRGC marker OPN4 and pan-RGC marker vGlut2-Sun1GFP reveal the levels of ipRGC survival. The red channel represents OPN4 staining, and the green channel represents the nucleus membrane GFP in RGCs. White circles with dash lines mark the ipRGCs. **m** Quantification of ipRGC survival rate under naloxone treatment in the NMDA model. $n = 4$ or 5 mice for each independent experimental condition. Panels **c**, **d**, **g**, **j**, and **m** data are presented as mean values ± SEM: unpaired $t$-test, two-tailed. The scale bar is 50 µm, which applies to all micrographs in the figure. Source data are provided as a Source Data file.

high-survival subclasses (ipRGC, αRGC, and Gpr88RGC) were alive at five days post-ONC, we did not observe further neuroprotection induced by Oprm1 on these three subclasses (Fig. 6e). In terms of the low-survival subclasses, the cell numbers significantly declined following ONC, but the overexpression of Oprm1 substantially reduced ONC-induced cell death, which indicated a transferable and broad protective effect of Oprm1 on all subclasses (Fig. 6e).

Finally, we deciphered the downstream signaling programs activated by the Oprm1 overexpression. By comparing the gene expression between the ONC sample and the Oprm1-overexpressed ONC sample, we identified 337 differentially expressed genes. Gene Ontology analysis (GO) revealed that the biological processes enhanced by Oprm1 overexpression were strongly associated with, though not restricted to, negative regulation of neuronal cell death, synaptic transmission, regulation of membrane potential, as well as neuron projection in the development and morphogenesis (Fig. 6f), which were in line with the observed neuroprotective effect of Oprm1 overexpression in multiple retinal injury models.

### Oprm1 maintains visual functions following RGC injury

We then investigated if the neuroprotection induced by Oprm1 improved visual function. First, we used the visual cliff test, which evaluates visual acuity based on the mouse's tendency to avoid visually deep areas. Since the ONC completely disrupts RGC projections to the brain, the visual cliff test was employed specifically with the NMDA model that did not completely damage the visual pathway. Mice were placed at the cliff area between shallow and deep sides (Fig. 6g), and the side selection by mice was recorded. We tested groups composed of five mice, each undergoing ten trials. Uninjured mice, on average, chose the shallow side 9.6 times out of 10 trials, which was in sharp contrast to the NMDA-treated group's average of 4.6 times (Fig. 6g). However, Oprm1 overexpression significantly improved the shallow side selection to 6.6 times on average in the NMDA-treated mice (Fig. 6g).

ipRGCs are the central relay for light signals to reach brain centers that regulate the pupillary light response (PLR). Therefore, we used the PLR to examine whether naloxone-induced ipRGC loss showed a physiological outcome on ipRGC signaling in mice. We employed sustained PLR as a marker to evaluate the status of ipRGCs in NMDA-injured retinas[40]. Although a decrease in the total number of ipRGC cells was observed after intravitreal injection of 20 mM NMDA (Fig. 5l and m), the resultant reduction in ipRGC numbers was not sufficient to induce PLR deficits (Fig. 6h), corroborating previous observations from Opn4aDTA mice with partial loss of ipRGC[41]. However, when the naloxone treatment was combined with the NMDA, a substantial reduction in ipRGC numbers was observed (Fig. 5l and m), and PLR deficits were detected, implying that the Oprm1-mediated ipRGC protection is attenuated by anti-opioid activity at the functional level.

### Protective effects of Oprm1 in a glaucoma model

Finally, we investigated whether overexpression of Oprm1 in RGCs could protect neurons from an experimental model of high intraocular

pressure-induced glaucoma. A previously established magnetic microbeads occlusion model[42] was used to increase the intraocular pressure (IOP). Specifically, the beads were intracamerally injected into the anterior chamber, adhering between the circumference of the cornea and iris and obstructing the circulation of aqueous humor (Fig. 7a). An elevation in IOP was then maintained for 8 weeks[3,42] (Fig. 7b). AAV2-mediated specific overexpression of Oprm1 in RGCs enhanced cell survival in glaucoma model at eight weeks after IOP elevation. The survival rate of RGC in the elevated IOP (eIOP) group was 62.2%, while the Oprm1 overexpression in RGCs significantly enhanced the survival rate to 84.0% (Fig. 7c and d, Supplementary Fig. 9a and b). Beyond the protection on somas of RGCs, the protective effect on optic nerve axon fibers was also assessed. We utilized toluidine blue staining to visualize myelinated nerve fibers. Cross-sections of the optic nerve, located 1 mm from the eyeball, were examined. The overexpression of Oprm1 resulted in a substantial increase in axon survival under eIOP, from 66.5% to 84.3% (Fig. 7e and f).

We used two visual-based behavioral assays to assess how visual functions in the glaucoma model were affected by Oprm1 overexpression at eight weeks following microbeads injection. We first tracked the free movement of mice in the visual cliff test arena over 30 min and analyzed their location distribution throughout this time. The uninjured sham group spent 73.6% of the time on the arena's shallow side, whereas the eIOP group mice spent only 46.5%, indicating reduced depth perception. Overexpression of Oprm1 in eIOP mice significantly increased the time on the shallow side to 63.4% (Fig. 7g and h). Second, we examined if the mice would retreat away from or pass through the shallow-deep boundary when moving from the shallow toward the deep side (Fig. 7i). The results showed that when entering the boundary zone from the shallow side, uninjured mice had a 59.9% chance to retreat. In comparison, the eIOP-injured mice only had a 10.3% chance to retreat. However, the eIOP mice with overexpressed Oprm1 had a 60.8% chance to retreat, comparable to that of uninjured mice (Fig. 7j). The results indicate that Oprm1 overexpression in RGCs improves the visual ability in the visual cliff test, likely through its role in neuroprotection.

## Discussion

### Organizing principles of cell−cell communication

This work established a systematic cell−cell communication atlas between RGCs and other retinal cell types in response to retinal injury. Our analysis has identified spatiotemporal organizing principles for cell−cell communications within a complex system composed of diverse cell types. First, we revealed that the numbers of ligand−receptor interactions between RGCs and other cell types varied based on the relative positions within the retina. Specifically, Müller glial cells and the astrocytes, which are in direct contact with RGCs, have the highest level of communication. In contrast, cell types physically separated from RGCs, such as cones and rods, have far fewer interactions. Second, we identified temporal variation in cell-cell

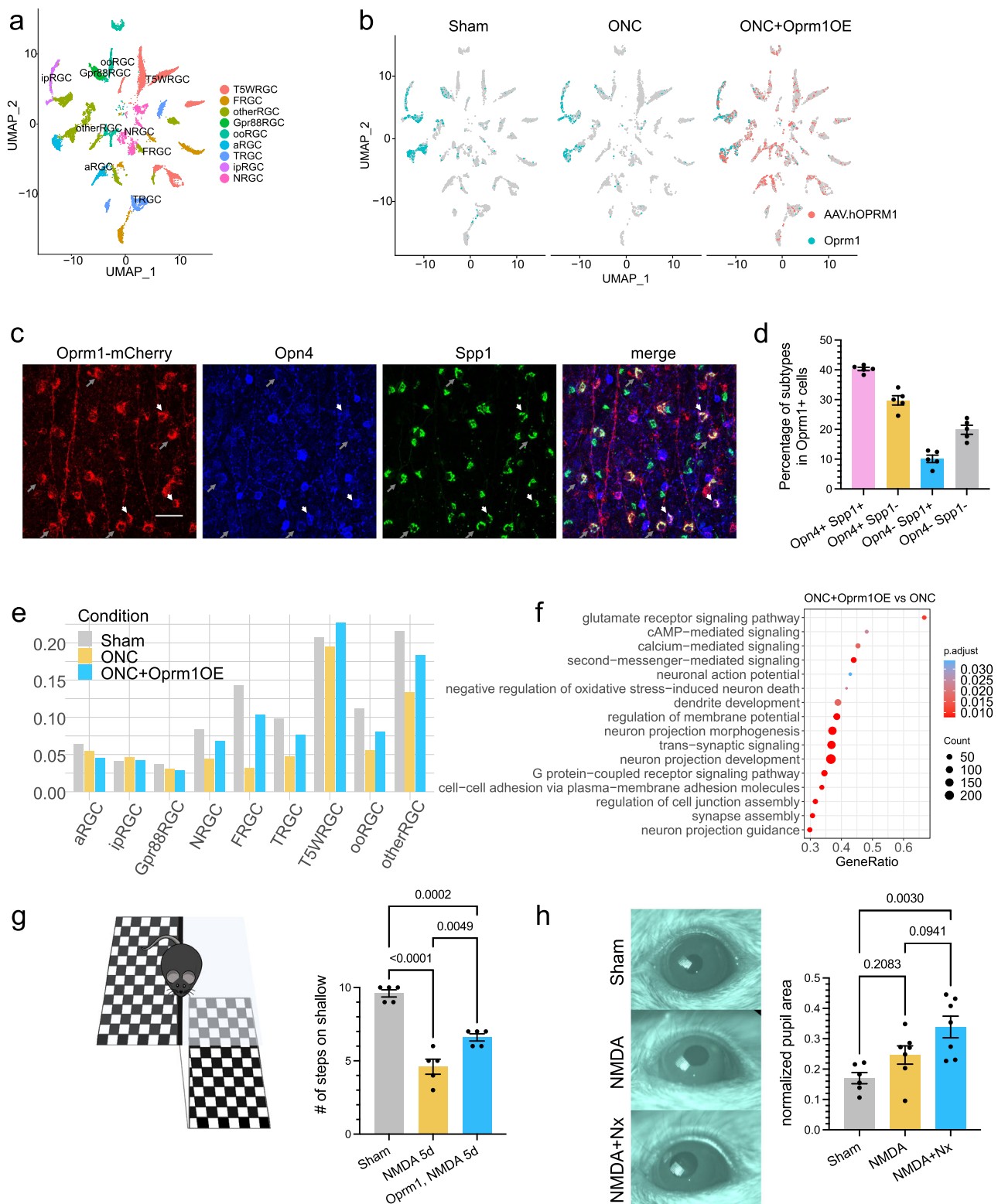

communication following optic nerve crush. Even though many cell-cell interactions were transient, returning to their original states quickly, we anticipate that the retinal cells received the distress signals from RGCs, and the response was already triggered in the short period of deviation from the original states. Finally, while the primary focus of our study is on the interactions between RGCs and other retinal cells, it's important to note that non-RGCs also engage in interactions with each other throughout this process (Supplementary Fig. 10a), contributing to the complexity of the interaction network.

## Cell-cell interactions as intrinsic cellular property

While the microenvironment was extensively studied in the cancer field, stem cell biology, and immunology, it has not been systematically analyzed in a mature nervous system. In this work, we identified neuroprotective factors from the perspective of cell-cell communication. One interesting finding is that the high-survival RGCs have more intercellular interactions than RGC subclasses with low survival abilities after injury. Furthermore, the fact that high-survival RGCs express more ligand and receptor genes suggests the property of resilience to

**Fig. 6 | snRNA-seq of RGCs after Oprm1 overexpression. a** Clustering of RGC subclasses profiled by snRNA-seq as an integration of cells collected from sham conditions, 5 days post-ONC, and the Oprm1-overexpressed 5 days post-ONC conditions. **b** UMAP shows the expression patterns of endogenous Oprm1 (cyan) and the ectopic expression of human Oprm1 (pink) in RGC subclasses. **c** Representative micrographs of whole-mount retinal tissues show the expression patterns of Opn4 and Spp1 in Oprm1-expressing RGCs. The red fluorescent staining is the mCherry tag in the Oprm1-mCherry reporter mouse strain; the blue represents Opn4 immunostaining, and the green represents Spp1 immunostaining. Grey long arrows point to the low-level Opn4-expressing cells; short white arrows indicate the medium-level Opn4-expressing cells. The scale bar is 50 μm. **d** Quantification of the ratio of Opn1 and Spp1 expression patterns in Oprm1-expressing RGCs. Data are presented as mean values ± SEM. $n = 5$ biological replicates of mice. **e** Normalized cell proportion among RGC subclasses under three conditions. **f** Representative biological processes enriched in Oprm1-

overexpressed five days post-ONC condition. Gene set enrichment analysis (GSEA) of Gene Ontology was adjusted using the Benjamini–Hochberg method. *p*-values for the GSEA test statistics are calculated by permutation (nPerm = 10,000). **g** Visual cliff test on the protective effect of Oprm1 overexpression against NMDA-induced vision impairment. Left: the equipment setup. Right: mean occurrences of mice selecting the shallow side (an indicator of acute vision) out of 10 trials. Each dot represents one mouse. Data are presented as mean values ± SEM. $n = 5$ mice for each independent experimental condition. Ordinary one-way ANOVA, Tukey's multiple comparisons test, with a single pooled variance. **h** Pupil reflex test on naloxone effect in NMDA model. Left: representative images of the pupil in three conditions. Right: normalized pupil size in three conditions. Data are presented as mean values ± SEM. $n = 6$ or 7 mice for each independent experimental condition. Ordinary one-way ANOVA, Tukey's multiple comparisons test, with a single pooled variance. Source data are provided as a Source Data file.

cell death is, in part, encoded within the cell–cell communication and established during development. Indeed, the concept of intrinsic mechanisms is not only constricted inside isolated cells. In a broad sense, communication with neighboring cells is another aspect of the intrinsic cellular property.

## All neighbors are good neighbors

Our analysis identified neuroprotective interactions from retinal cells to RGCs by comparing the high- and low-survival RGC subclasses. It was unexpected that we only found pro-RGC survival interactions stronger in high-survival than in low-survival RGCs. We did not identify interactions sent from retinal cells to the low-survival RGCs preferentially, which likely represented the neurotoxic interactions. In other words, the neighboring retinal cells are helping to promote the survival of a subset of RGCs post-injury rather than sending any additional apoptotic signals to the low-survival RGC subclasses. This starkly contrasts the transcriptome comparison among RGC subclasses, in which the optic nerve crush operation induces significantly differentially expressed genes in either high- or low-survival subclasses of RGCs. It is worth noting that we performed our analysis only up to two days post-ONC when almost all RGCs remained alive. Therefore, it is unlikely that RGC death prevented us from detecting neurotoxic interactions.

## An Opioid receptor with a moonlighting function

We functionally validated one of the predicted neuroprotective factors identified in this analysis - the μ opioid receptor encoded by the *Oprm1* gene. Despite its well-documented role in pain regulation, reward, and addictive behaviors[17–19], it has not been previously implicated in regulating neuronal survival. Both gain- and loss-of-function studies demonstrated that Oprm1 is necessary and sufficient to promote RGC survival following multiple types of injury (Supplementary Fig. 10b). The detailed analysis of Oprm1 expression patterns in ipRGCs and αRGCs revealed a high expression level within the major subtypes of ipRGCs and αON-S/M4 subtype (Supplementary Fig. 10c–e), which are highly consistent with their enhanced survival capabilities[4,28,29]. Although endogenous Oprm1 is predominantly expressed in subtypes of ipRGCs, Oprm1 overexpression in pan-RGCs can also protect more vulnerable RGC subtypes. Oprm1 overexpression in RGCs significantly improved visual perception in a high intraocular pressure-induced glaucoma model, implying its future translational potential. Interestingly, Oprm1 is also predominately expressed in human ipRGC[43], suggesting activating the Oprm1 receptor represents a potential therapeutic approach. Therefore, the high-affinity selective agonists for Oprm1 that are already in clinical use for pain management may be repurposed for RGC neuroprotection in emergencies following acute optic nerve injury or in closed-angle glaucoma. Furthermore, exploring the potential protective effects of Oprm1 on other CNS neurons could be an intriguing topic for future research.

Together, our study established an effective strategy to discover functionally important cell–cell communication in a complex in vivo system. Moreover, the combined experimental and analytic platform will facilitate the mechanistic study of intercellular interactions in diverse biological systems beyond the nervous system.

## Methods
### Mice
All animal experiments were approved by the Institutional Animal Care and Use Committees (IACUC) at Johns Hopkins University School of Medicine. The mice were maintained in a climate and light/dark cycle (14 h/10 h) controlled pathogen-free facility, with continuous access to food and water. All studies were conducted on adult mice aged 2–3 months. Both male and female mice were included in the study. The following mouse strains were utilized: C57BL/6 mice (JAX #000664), vGlut2-IRES-Cre (JAX #016963), CAG-LSL-Sun1/sfGFP mice (JAX #021039), and MOR-mCherry reporter mice (JAX #029013), which were all available at the Jackson Laboratory (Bar Harbor, ME). The homozygotes vGlut2-Cre; LSL-Sun1GFP mice were generated by breeding and backcross. All surgeries and experiments on the live animals were performed under anesthesia induced by intraperitoneal injection of Avertin solution (300 mg/kg, diluted in sterile PBS). Details are described in each specific technical section.

### Optic nerve crush (ONC)
After anesthesia with Avertin solution (300 mg/kg intraperitoneal injection), the right optic nerve was exposed infraorbital and crushed with #5 Dumont forceps (FST, Foster City, CA) gently at ~1 mm behind the eyeball. The sham operation was conducted with the same anesthesia procedure and optic nerve exposure under the microscope without crushing the optic nerve. All microsurgeries were performed by experienced personnel using the Leica M80 stereo microscope.

### Retinal tissue dissociation and droplet-based scRNA-seq
With 12 h, 1-, 2-day-ONC, or under the sham conditions, the mice were euthanized following anesthesia induced by intraperitoneal injection of Avertin solution. Each time point had two biological replicates of mice. The eyeballs were enucleated and kept in ice-cold PBS. The retinas were quickly dissected, and whole retinal cells were dissociated using the Papain enzyme system (LK003150, Worthington). 30 μM actinomycin-D was added to the Papain system to prevent post-enucleation transcription. The tissues then underwent manual trituration in a neurobasal medium supplemented with 3% BSA, 10 mg/mL Ovomucoid, 1 U/μL RNase inhibitor, and 3 μM actinomycin-D. Cells were filtered through a 40-μm strainer and centrifuged under $500 \times g$. For scRNA-seq, the cell concentration was kept at ~1000 cells/μL. The scRNA-seq libraries were prepared using the 10X Genomics Chromium 3' Single-Cell Gene Expression V3 Kit following the manufacturer's

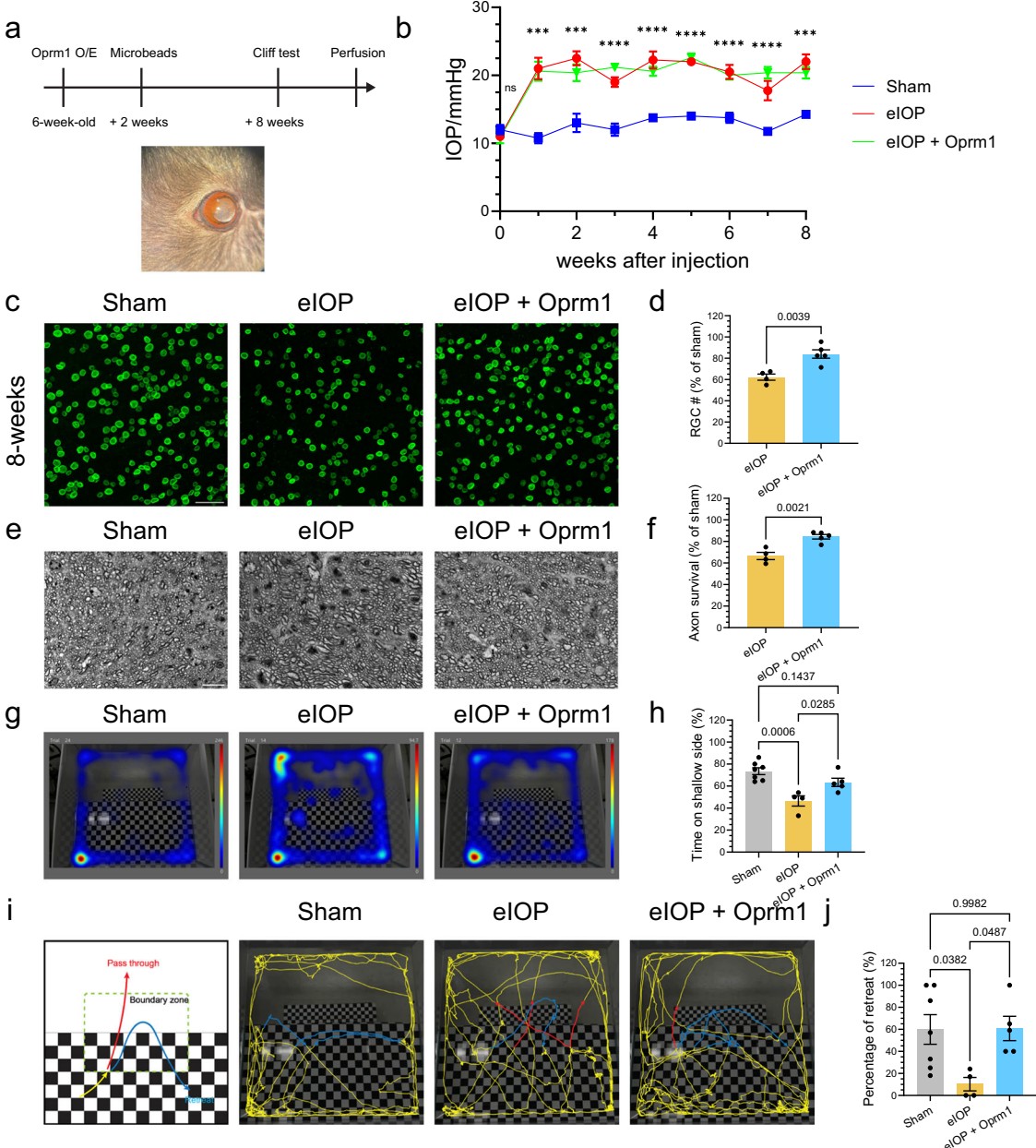

**Fig. 7 | Oprm1 effect in a glaucoma model. a** Experimental model for the elevation of intraocular pressure (IOP) induced by intracameral injection of magnetic microbeads. A representative photo shows magnetic microbeads stuck between the circumference of the cornea and iris after injection. **b** IOP values across 8 weeks. Data are presented as mean ± SEM. Sham group, *n* = 4 mice; eIOP group, *n* = 4 mice; Oprm1 + eIOP, *n* = 5 mice. One-way ANOVA. *p*-values: 0.4588 (wk 0), 0.0004 (wk 1), 0.0009 (wk2), <0.0001 (wk 3), <0.0001 (wk 4), <0.0001 (wk 5), <0.0001 (wk 6), <0.0001 (wk 7), 0.0002 (wk 8). **c** and **d** Protection of RGCs from eIOP with over-expression of Oprm1. Data are presented as mean values ± SEM. *n* = 4 or 5 mice for each independent experimental condition. **e** protection of axons in optic nerve with overexpression of Oprm1. **f** Quantification of axon survival, shown as per-centages of axons relative to those in the sham group. Data are presented as mean values ± SEM. *n* = 4 or 5 mice for each independent experimental condition. **g**–**j** Assessment of Oprm1 overexpression on visual function using the vision cliff arena recording. **g** Representative movement heatmaps from sham, eIOP, and Oprm1+eIOP. **h** Quantification of time spent on the shallow side during the 30 minutes in the test. **i** A schematic diagram and representative traces (5 min) of the visual cliff test. Red traces highlight the pass-through from the shallow to the deep side, while blue traces highlight the retreat to the shallow side. **j** Percentages of retreat events among all traces entering the boundary zone during the 30 min. Data are presented as mean ± SEM. Sham group, *n* = 7 mice; eIOP group, *n* = 4 mice; Oprm1 + eIOP, *n* = 5 mice. One-way ANOVA, multiple comparisons. Scale bars: 50 μm (**c**), 10 μm (**e**). Source data are provided as a Source Data file.

---

standard protocol. Approximately 8000–12,000 cells were loaded. Libraries were sequenced with the NovaSeq S2 100 platform.

### Quality control, clustering, and cell type identification of scRNA-seq data

Raw reads of adult mouse retina scRNA-seq data before and post-ONC (control, 12 h, 1, and 2 days) were mapped to mm10, and expression matrices were generated using Cell Ranger 3.0.2 from 10x Genomics. In addition, we downloaded the corresponding RGC data from GEO (GSE137400). We created a Seurat object for each sample using Seurat v4.0.5. For the whole retina dataset, cells were removed if their nUMI was <800 or >30,000, nGene was <350 or >7500, mitochondrial gene rate was >20%, or log10GenesPerUMI was <0.8. For the RGC dataset, we removed cells with nUMI <500, nGene <250, mitochondrial rate

>20%, or log10GenesPerUMI < 0.8. We then used Scrublet v0.2.3 to remove predicted doublets with default parameters for both data sets. Furthermore, genes detected in less than five cells in each sample were also removed from the data. As a result, 56,531 cells from the whole retina dataset and 86,426 cells from the RGC dataset were obtained for downstream analysis. For each dataset, Seurat objects of the samples were then normalized, integrated, clustered, and visualized through UMAP using Seurat v4.0.5. Cell identities were assigned based on retinal cell types' known canonical marker genes.

## Magnetic purification of RGC nuclei for droplet-based snRNA-seq

The vGlut2-Cre; LSL-Sun1GFP mice were used for this experiment. Mice under sham condition, post-ONC[5d], and post-ONC[5d] with hOprm1 overexpression conditions were then euthanized under anesthesia induced by intraperitoneal injection of Avertin solution. The retinas were quickly and carefully dissected out from enucleated eyeballs. Through gentle pipette trituration, the retinas were dissociated in an ice-cold neurobasal medium on ice. According to the 10X Genomics protocol, whole retinal cells were then incubated in the nuclear lysis buffer on ice for 3 min to expose and penetrate the cell nuclei and then centrifuged to pellet the nuclei. The cell nuclei were then resuspended in ice-cold MACS buffer (Miltenyi Biotec, Gaithersburg, MD) for anti-GFP MACS microbeads incubation. The whole retina nuclei were then passed through and eluted from the Miltenyi MACS MS column to enrich GFP + RGC nuclei and then centrifuged into the pellet. The pellet was then resuspended to reach a 4000–5000 nuclei/μL loading concentration onto the 10X Genomics Single-Nuclei 3' HT platform.

## Data analysis of snRNA-seq of purified RGCs

For purified RGC snRNA-seq data from sham, 5-day post-ONC, and 5-day post-ONC with Oprm1 overexpression samples, raw reads were mapped to mm10, and expression matrices were generated using Cell Ranger 7.0.0 from 10x Genomics. A Seurat object for each sample was created using Seurat v4.3.0. For sham data, cells were removed if nUMI was <1000 or >50,000, nGene was <500 or >10,000, the mitochondrial rate was >10%, or log10GenesPerUMI was <0.8. For ONC and ONC with *Oprm1* overexpression data, cells were removed if nUMI was <1000 or >30,000, nGene was <500 or >7000, the mitochondrial rate was >10%, or log10GenesPerUMI was <0.8. We then removed predicted doublets using DoubletFinder v2.0.3 with its standard pipeline for each sample[44]. In total, 26,170 cells were used for downstream analysis. Specifically, 9306, 6060, and 10,804 cells were obtained from the sham, ONC, and ONC with Oprm1 overexpression samples, respectively. The filtered Seurat objects were normalized, integrated, clustered, and visualized through UMAP using Seurat v4.3.0. Cell type identities were assigned based on retinal cell types' known canonical marker genes. After cell type annotation, 15,803 RGCs were obtained, with 6348, 3314, and 6141 RGCs from the sham, ONC, and ONC with Oprm1 overexpression samples, respectively. We then took the subset of the Seurat objects of each sample by removing non-RGCs and performed normalization, integration, clustering, and UMAP visualization again using Seurat. RGC subtype identities were then assigned based on the projection of the RGC atlas data from Tran et al.[4] onto our clusters using the Seurat data transfer pipeline and known subtype marker genes.

## Prediction of ligand–receptor interactions between RGCs and other cells

We predicted ligand–receptor interactions between RGCs and other retinal cells and between each RGC subclass and other cells using the package LRLoop. The details of the algorithm can be found in ref. 23. In essence, the LRLoop utilized three major types of information to predict the ligand-receptor interactions: (1) The gene expression level of ligands in sender cells and receptors in receiver cells; (2) The cell-type-specific signaling and regulatory networks by superimposing the known networks and the expression levels of the genes in signaling and regulatory networks in a particular cell type; (3) The ligand–receptor interaction pairs that form a loop (examples in Fig. 2e and Supplementary Fig. 2d) through the signaling and regulatory networks. We first derived the interaction strength of each ligand–receptor interaction using the first two pieces of information. We then modified the strength by considering the other ligand–receptor interactions that can form a loop with the interaction of interest. Finally, the candidate ligand–receptor interactions are further filtered based on the following criteria: (1) both the ligand and the receptor genes are detected in at least 10% of the corresponding sender and receiver cell types, respectively; (2) The interaction score calculated from LRLoop is no <0.5 in any time point. The overall score between each pair of cell types was defined as the sum of interaction scores of identified ligand–receptor pairs.

## Differential ligand–receptor interactions between high- and low-survival RGC subclasses

For the predicted ligand–receptor interactions from each non-RGC retinal cell-type ct1 to RGC subclasses, DSLR, the interaction score difference of each LR pair between high and low survival RGC subclasses was defined as

$$DS_{LR} = mean\left\{\max_{time\,points}(S_{LR}): from\ ct1\ to\ highRGC\right\}$$
$$- mean\left\{\max_{timepoints}(S_{LR}): from\ ct1\ to\ lowRGC\right\}$$

Interactions with score differences above or below cutoffs (0.25 and −0.25) were considered differential ligand–receptor interactions between high- and low-survival RGC subclasses.

## Fluorescence in situ hybridization (FISH)

Eyes with optic nerves were surgically removed from perfused mice following anesthesia induced by intraperitoneal injection of Avertin solution, and retinas were dissected out and post-fixed in 4% PFA at 4 °C overnight. After a serial transfer in 10%, 20%, and 30% sucrose solution, dehydrated retinas were embedded in O.C.T. compound (REF 4583, Sakura) for cryosection into 10 μm cross-sections. According to the manufacturer's instructions, fluorescent in situ hybridization (FISH) was performed using the commercially available RNAscope Fluorescent Multiplex assay (ACDbio). In brief, the O.C.T. compound was removed by washing with 1×PBS, followed by baking the slides for 30 min at 60 °C. Sample slides were then post-fixed with 4% PFA for 15 min at 4 °C and serially dehydrated with 50%, 70%, and 100% ethanol for 5 min at room temperature. Samples were covered with hydrogen peroxide for 10 min and then boiled in 1 × Target Retrieval Reagents for 5 min). The samples were incubated with Protease III in the HybEZ™ II Hybridization System for 30 min at 40 °C. After incubation with RNAscope probes for 2 h at 40 °C and subsequent AMP), the samples were sequentially applied with HRP, Opal fluorescent dyes, and HRP Blocker for corresponding channels. The sample slides were mounted with Fluoroshield and coverslips. Confocal images of the slides were acquired using a Zeiss LSM 880 microscope.

## Quantification of fluorescence in situ hybridization

Confocal images were acquired using a Zeiss LSM 880 microscope (JHU Institute for Basic Biomedical Sciences Microscope Facility) under the same parameter settings. In addition to DAPI, four microscope channels were used: Fluorescein for Opal520 (FP1487001KT, Akoya Biosciences) to show ligand expression (Manual Assay RNAscope Multiplex Probe for Anxa1, 509291-C1, ACD Bio), Cyanine 3 for Opal570 (FP1488001KT) to show receptor expression (Dysf, 1134891-C2), Texas Red for Opal620 (FP1495001KT) to show Müller glia with a

probe against Slc1a3 (430781-C3), Cyanine 5.5 for Opal690 (FP1497001KT) to show αRGCs with a probe against Spp1 (435191-C4).

Quantification of fluorescence in situ hybridization was obtained based on the guidelines provided by the RNAscope Fluorescent Multiplex assay (ACDbio). In brief, a threshold was set to define the shape of each discrete signal dot after the average background intensity was calculated. The average intensity per single dot was calculated within a representative region containing around 20 signal dots by the following equation:

$$\text{Average intensity per single dot} = \frac{\sum \text{Integrated intensity of dots} - \text{Average background intensity} \times \sum \text{Area of dots}}{\text{Number of dots}}$$

Then, the total dot number in a region above the threshold was calculated by the following equation:

$$\text{Total dot number} = \frac{\text{Total intensity} - \text{Average background intensity} \times \text{Total area}}{\text{Average intensity per single dot}}$$

### Intravitreal injection of AAV2, NMDA, and intraperitoneal injection of morphine

The intravitreal injection of the AAV2 virus was performed as previously described[45]. Under anesthesia following anesthesia induced by intraperitoneal injection of Avertin solution, 1.5 μL of AAV2 virus was injected into the right vitreous humor of a mouse with a Hamilton syringe assembled with a 32-gauge needle. The position and direction of the injection were well-controlled to avoid injuring the lens. AAV2-Syn1-FLEX-hOprm1-mCherry (Addgene plasmid #166970) virus or AAV2-FLEX-sun1GFP (Addgene plasmid #160141) control virus was packaged by SignaGen at high-titer ($1 \times 10^{13}$ gc/mL). For the NMDA excitotoxicity model, the mice were intravitreally injected with 1.5 μL of DMSO-dissolved 20 mM N-methyl-D-aspartic acid (NMDA) (Sigma-Aldrich, St. Louis, MO). The commercially available morphine solution at 1.0 mg/mL in methanol (M-005-1ML, Sigma Aldrich, MO) was diluted with PBS for intraperitoneal injection into mice to reach 0.5 mg/kg final concentration.

### Immunohistochemistry and microscopy

Eyeballs were surgically enucleated from 4% PFA-perfused mice following anesthesia induced by Avertin. The eyeballs were post-fixed in the 4% PFA at 4 °C overnight. Retinas were dissected in 1 × PBS and cut with four radial incisions to create a petal shape. Cold (−20 °C) methanol was used to fix and flatten the retinas. Retinas were transferred into a 24-well plate with cold methanol for storage[46]. For immunohistochemistry, retinas were blocked in a blocking solution containing 0.3% Triton X-100, 0.2% BSA, and 5% goat or donkey serum accordingly in PBS at room temperature for 1 h, followed by incubation with the primary antibodies at 4 °C overnight, after washing 4 × 10 min in 1 × PBS with 0.3% Triton X-100, retinas were incubated with secondary antibodies at room temperature for 4 h. Finally, the retinas were washed for 4 × 15 min in 1 × PBS with 0.3% Triton X-100 and mounted with the Fluoroshield (F6182, Sigma Aldrich). The following primary antibodies were used accordingly: rabbit polyclonal anti-RBPMS (Invitrogen, PA5-31231, 1:500), chicken polyclonal anti-GFP (Invitrogen, A10262, 1:500), rabbit polyclonal anti-OPRM1 (Invitrogen, PA1-24628, 1:500), rabbit polyclonal anti-Opn4 (Invitrogen, PA1-780, 1:500) for Supplementary Fig. 9, rabbit polyclonal anti-Melanopsin (AB-N38, Advanced Targeting Systems, 1:500), rabbit mAb anti-Tbr1(CST #49661, 1:200), rabbit mAb anti-Satb1 (abcam ab109122, 1:200), rabbit polyclonal anti-Foxp2 (abcam ab16046, 1:500), goat polyclonal IgG anti-Osteopontin (biotechne AF808, diluted to 5ug/mL). For staining the mCherry tag of the MOR-mCherry reporter mice (JAX #029013) or the AAV2-FLEX-hOprm1-mCherry virus-infected samples, a rat mAb anti-mCherry was used (Invitrogen, M11217,

1:500). Following secondary antibodies were used correspondingly: Alexa Fluor 647 goat anti-rabbit IgG (Invitrogen A21245, 1:500), Alexa Fluor 488 goat anti-chicken IgY (Invitrogen A11039, 1:500), Alexa Fluor 568 goat anti-rabbit IgG (Invitrogen A11011, 1:500), Alexa Fluor 568 goat anti-rat IgG (Invitrogen A11077, 1:500), donkey anti-Goat IgG Alexa Fluor Plus 647 (Thermo A32849TR, 1:500), donkey anti-Rat Alexa Fluor 594 (Thermo A21209, 1:500), donkey anti-Rabbit IgG Alexa Fluor 488 (Thermo A21206, 1:500), donkey anti-Chicken IgY Alexa Fluor 488 (Thermo A78948, 1:500).

Confocal images were acquired using a Zeiss LSM 800 microscope (JHU neuroscience MPI core) for RGC numbers counting, eight squares (320 μm × 320 μm) were sampled evenly around the peripheral region of each retina. The mean cell number of these eight positions was viewed as the RGC cell number of one retina as one biological replicate.

### Elevation of intraocular pressure (IOP) by intracameral injection of microbeads

Elevation of IOP was induced by intracameral injection of magnetic microbeads into the right eye of the VGlut2-Cre; LSL-Sun1-sfGFP mice, under anesthesia, induced by intraperitoneal injection of Avertin solution. In brief, superparamagnetic polystyrene beads (4.5 μm, Dynabeads® M-450 Epoxy, Invitrogen, 14011) were treated with 0.02 M sodium hydroxide (NaOH) in 10 × Tris buffer to remove epoxy groups. Then, the microbeads were homogenized in a sterile balanced salt solution at a concentration of $1.6 \times 10^6$ beads/μL. Three-month-old mice were anesthetized with an intraperitoneal injection of proper volume of Avertin (2,2,2-Tribromoethanol, T48402, Sigma Aldrich, 20 mg/mL, 250 mg/kg). After pupil dilation with a 1% tropicamide ophthalmic solution (NDC 70069-121-01, Somerset Therapeutics LLC, Hollywood, FL), a 30 G × ½ needle (REF 305106, BD PrecisionGlide™) was used to incise the edge of the cornea, and 1.5 μL of microbeads solution was injected into the anterior chamber using a sharp glass micropipette connected to a Hamilton syringe (Cat # 20919, World Precision Instruments). A small piece of the magnetic ring was applied to evenly distribute microbeads around the circumference between the cornea and iris[42].

### IOP measurement

The IOP of both eyes was measured with the TonoLab tonometer (Colonial Medical Supply) every week. The probe tip was applied 1–4 mm from the center of the cornea after the mice were anesthetized. The IOP on each eye was measured six times, and the tonometer produced an average value of IOP automatically. All IOPs were measured around 4:00–5:00 pm to reduce variation.

### Visual cliff test

Mice were put into a transparent plexiglass arena (Conduct Science, Boston, MA) with a dimension of 62 cm × 62 cm × 19 cm ($L \times W \times H$). The arena was evenly separated into two zones: a shallow zone with a checkered pattern under it and a deep zone with the same checkered pattern 60 cm underneath it to create an illusion of depth (left panel in Fig. 6g). NMDA-treated mice were placed in the middle line, and their choices toward different zones were recorded. The arena was cleaned between tests. For the elevated IOP glaucoma model, mice did not show significant differences in zone choices between groups two weeks after IOP elevation. Eight weeks after IOP elevation, the mice tended to stay at the starting point without much movement when placed in the arena. This may have been caused by the weekly anesthesia with Isoflurane for IOP measurement. Such behavior is consistent with previous reports showing that repetitive applications of Isoflurane could lead to cognition impairment[47,48]. For these mice, 30-min movement videos were recorded after a 5-min adaptation time. EthoVision XT 17 (Noduls Information Technology Inc.) was used to record and analyze videos.

Trace analysis was performed based on the videos recording mice locomotion. Two types of behavior were analyzed. First, we calculated the time the mice stayed in either the shallow or the deep zone. We then analyzed the movement in the boundary zone in the center of the arena because the mice tended to move in a circulating way along the borders but showed different behaviors when in the center. The boundary zone was defined as a rectangle extending three grids into either the shallow or deep sides from the middle line and three away from the lateral borders of the area (the length of 3 grids is nearly the average body length of the mice without tails, Fig. 7i). The traces of mice entering the boundary zone from the shallow side were counted. If mice leave the boundary zone to the deep side, the trace is counted as a pass-through. If mice leave from the zone to the shallow side, it is estimated as a retreat. The ratio of retreats to total traces was calculated.

### Intraperitoneal injection of Naloxone and ipRGC number analysis

Six-week-old vGlut2-Cre; Sun1-GFP mice were used in experiments related to Naloxone. For the NMDA damage model, 0.4 mL of 0.5 mg/mL Naloxone (0.2 mg/mouse)[49,50] was intraperitoneally injected daily, and mice were sacrificed 4 days after NMDA damage. For the ONC model, 0.4 mL of 0.5 mg/mL Naloxone was intraperitoneally injected daily, and mice were sacrificed five days after injury. Opn4 antibodies (PA1-780, Invitrogen) were used to label ipRGCs in immunostaining. Photos of the whole mounted retina were taken with a Zeiss LSM 880 Confocal Microscope. Numbers of ipRGC in each retina were counted.

### Pupillometry

Mice for pupillometry were 8-week-old male B6129SF1/J wildtype from Jackson Laboratory. The basic procedure for pupillary light reflex (PLR) recording and pupil size quantification was described in Keenan et al.[40]. Mice were dark-adapted for 2 h before the PLR experiment. They were briefly restrained by hand for video recording. Two short videos were shot for each mouse, one for dark-adapted baseline pupil size as a reference point and the other for pupil constriction measurement after 30 min light exposure, mainly attributed to melanopsin phototransduction in ipRGCs. A Sony Handycam (FDR-AX33, prime lens, manual focusing, night shot mode, external infrared light source) was used for recording. The dotted pattern from the infrared light source is reflected by the mouse's cornea and used as the focusing indicator. This setup ensures that the mouse's pupils can only be focused on a fixed distance from the camera and can be easily visualized in the dark. Light at an intensity of 0.327 W/m$^2$ was used to induce PLR. Using Fiji software, pupil size was measured as the maximum diameter in each picture frame. The pupil constriction is quantified by normalizing the pupil area at 30 min to the baseline area in the dark. A one-way ANOVA statistical analysis was performed.

### Axon protection examination

To examine Oprm1 protection on axon fibers in the glaucoma model, optic nerves were fixed in 2.5% glutaraldehyde and 2% paraformaldehyde in 0.1 M sodium cacodylate buffer overnight at 4 °C. After being embedded in resin, 1 µm thick sections of the optic nerve 1 mm distant from the eyeball were collected with a vibratome and stained with toluidine blue. Bright-field images from areas in the peripheral region were collected by a Zeiss AXIO microscope with Neurolucida under a ×100 lens. Three to four areas were imaged from each section, and three to five optic nerve sections were examined in each sample. Images were first analyzed with the CLAHE plugin in ImageJ. Then, the number of axons was calculated with the particle analysis function after being adjusted with the auto-local threshold function. The sizes were set from 0.005 to 0.05, and the circularity was from 0.1 to 1.

### Reporting summary

Further information on research design is available in the Nature Portfolio Reporting Summary linked to this article.

## Data availability

The scRNA-seq and snRNA-seq data generated in this study have been deposited in the GEO database under accession code GSE241268, GSE248537, and GSE248868. The scRNA-seq data generated in this study can also be visualized at the Broad Institute Single Cell portal under accession code SCP2423. Source data are provided with this paper.

## Code availability

Computational programming codes for ligand-receptor pairs analyses are stored in GitHub and figshare, https://github.com/Pinlyu3/LRLoop; https://doi.org/10.6084/m9.figshare.20126138.v1.

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

## Acknowledgements

This work was supported by the National Institutes of Health (5R01EY031779 to J.Q. and F.Z.). We are grateful to Drs. Harry Quigley, Elia Duh, Pingwu Zhang, and the members of Qian's lab for comments and suggestions. We highly appreciate Linda D. Orzolek and Tyler Creamer at the Johns Hopkins Single Cell and Transcriptomics Core for sc/snRNA-seq. We also appreciate the Multiphoton Imaging Core (MPI) in the Solomon H. Snyder Department of Neuroscience at the Johns Hopkins (5P30NS050274), as well as the Wilmer Eye Institute imaging core at the Johns Hopkins (5P30EY001765). We thank Yiwen Yan for her participation in quantifying cell survival and mouse behavioral tests.

## Author contributions

C. Qian., Y.X., F.-Q.Z., and J.Q. conceptualized the project and designed experiments. C. Qian, Y.X., C. Qi, H.W. and B.D. conducted experiments. Y.X. mapped the sequencing data and performed the bioinformatic analyses. C. Qian, Y.X., F.-Q.Z., and J.Q. wrote the manuscript with inputs from all authors. D.J.Z., S.H., and S.B. helped interpret the data and provided reagents.

## Competing interests

The authors declare no competing interests.
