## [Peer Review File · Nature Communications]

Intercellular communication atlas reveals Oprm1 as a neuroprotective factor for retinal ganglion cellsREVIEWER COMMENTS

Reviewer #1 (Remarks to the Author):

The current study by Qian J. and Zhou F. systematically surveyed cell-cell interactions among neurons and non-neuronal cells in the retina. The work was put in a degeneration (ONC and glaucoma) setting, and the authors provided a systematic survey using their own algorithm. The bioinformatic prediction landed on an interesting hit named *Oprm1*, enriched in ipRGCs, but has a broad impact on neuroprotection in both injuries and degeneration models.

Overall, the work is innovative, thorough, and interdisciplinary. It provides a model for others in retina neurobiology and neurodegeneration to study biological processes in the cell-cell interaction. This current study conceptually advances the field, rather than the past, focusing on RGC-intrinsic properties. The work is timely, given the ever-exploding scRNA-seq data – the authors combined multidisciplinary approaches and executed a beautiful storyline from in silico prediction to in vivo translational research. The team carried out multiple collaborative experiments at the Hopkins Med school, which was truly impressive as a dream team. Thus, the study would greatly interest general readers in neuroscience, ophthalmology, neurology, and regenerative medicine. The application of innovation of the bioinformatics approaches was also a huge plus for many others to follow.

While the authors focused on *Oprm1* biology, I view this paper more like a method/resource for the field to share, as they also highly cited Tran 2019, *Neuron*. Thus, my major points were listed below but largely on the organization of the existing data, the transparency of the report, as well as the upcoming data sharing/database look-up for general readers to learn and utilize in the field. Specific points listed below would benefit the general readers in the published format.

Major Points:

- 1) The biggest highlight of this paper is a systematic survey of all known ligand-receptor interactions: (a) for clarity for all readers, especially those outside the bioinformatics field, the authors may want to consider organizing the Ligand-Receptor (L to R) and keep it highly consistent with the figures/text, and also make it very clear to the readers – I am often lost by reading this; (b) a central hypothesis here is that RGCs provide the ligands but the other cells serve as receptors. – could the authors elaborate the data analysis in a reverse manner regarding the potential role of ligands from non-retinal neurons onto RGCs (receptors)? It might be beyond the scope of the current paper, but it would be a complementary resource for the field to understand this intriguing question and emerging field. (c) Are more time points offered in the current study – does that change the interactome? Esp. the Tran paper offered multiple time-points – such as 3-4dpc? 1wpc?
- 2) The set of images within Fig 3d provided a solid experimental basis for the bioinformatics prediction – however, the current set of images was very hard to read and interpret as four-color images – The authors should consider providing high-resolution and split 4-channel images in a dedicated supplementary figure.
- 3) Interpretation for Fig 4e raises some questions – the authors need to consider the distance, especially as all RGCs are packed together, providing immediate access to the neighbors – thus, even though the ligand expression level is low, the paracrine effects within RGC groups can be prominent for neuroprotection – these are interesting and unexplored areas which the authors provided the first insights- yet the conclusion and discussion on these points need to be carefully flashed out.
- 4) The transition from Fig 5 to the top of Fig 6 needs more experimental data to verify the findings = Using the same preparations as the authors did in Fig 5l/m with only anti-opn4 staining, the authors should consider integrating more antibody staining (rather than only focusing on *Opn4*./ipRGCs) – this is a critical piece of data validating the finding. The antibodies should be included to verify the findings include
goat anti-Osteopontin/Spp1 R&D Systems Cat#AF808; rabbit anti-Melanopsin ATSBio Cat#AB-N38;

rabbit anti-Foxp2 Abcam Cat#ab16046; rabbit anti-Tbr1 Cell Signaling Technology Cat#49661; rabbit anti-Satb1 Abcam Cat#ab109122;

5) The entire Figure 5 is very critical for the manuscript. However, I am confused about the random data points that the authors chose to analyze the end-points – Fig 5b included 14dpi; Figure 5f included 7dpi; Figure I was 5dpi; Figure L was 4dpi – I understand that they were different models – but some systematic organization and comparison, and explanations to the readers/reviewers are highly desired here.

6) Related to Major Point 4 – it would be critical for the authors to carry out double-color immunostaining for Oprm1 [one antibody was applied in Fig 7b, but I am not too sure about the specificity] onto Spp1, Opn4, Tbr1, foxp2, etc? (alternatively, double-color in situ/RNA-scope for Oprm1 onto several markers) – The expression level/percentages must be quantified to further support the findings in both Fig 5/ Fig 6a-c.

7) Related to Fig S7 and the paper in general: The method section did NOT include the antibody information (Catalog, species) for immunohistochemistry and their dilutions unless the information was included somewhere else in the report summary that I neglected.

8) Related to Major Point 1 - In the revised format, the authors would need to lay out their plan to share the big data interactively to lay audience and non-bioinformatics experts – some samples from Tran et al., 2019 at the Broad Institute would be a good example.

https://singlecell.broadinstitute.org/single_cell/study/SCP509/mouse-retinal-ganglion-cell-adult-atlas-and-optic-nerve-crush-time-series/

The authors are leading experts in bioinformatics and, thus, should have the bandwidth to share such data with the public in compliance with the NIH Data sharing rules.

MINOR Points

1) Discussion: The study performed their analysis at 2 weeks following optic nerve crush, at which 50-80% RGCs are already lost. This limits the interpretation of the cell-cell interaction data, for example. The statement that "all neighbors are good neighbors" and that neighboring RGCs are helping to promote RGC survival "rather than sending any additional apoptotic signals to destroy injured RGCs" is somewhat misleading- this could be due to the study focusing only on the few surviving cells after 2 weeks. Two weeks could be too long to see negative interactions. If data are available, maybe consider additional time points (days and 1-week post crush) to see how cell-cell interactions change as RGCs are lost would be extremely informative.

2) PMID: 22661469 should be considered a citation relevant to the concept here.

3) Line 175 citations - the citation of Ref 32 should be added to 4&27; not sure about Ref 28 as it was a zebrafish paper - unless the authors utilized its data – I would delete it.

4) Line 221 citations are missing Ref 27 and PMID: 37624696.

5) The readers would benefit from an illustrative model describing the biology of Oprm1 in terms of the RGCs' neuroprotection.

Reviewer #2 (Remarks to the Author):

Summary

Using scRNA-seq with mouse retinal cells before and after optic nerve crush, this group focuses on the potential interaction of glial cells with RGCs mediated by cell surface ligands and receptors. Bioinformatic analysis of transcriptomes of non-RGC retinal cells collected by this group and previously published RGCs, they identified a group of ligand-receptor pairs seems upregulated in surviving RGCs after axotomy. They continue to demonstrate that one of the receptors, Oprm1, showed neuroprotection in three optic neuropathy models.

Although the finding is interesting and intriguing, the data analysis is overly simplified, especially lacking RGCs from the same batch of non-RGC collections jeopardize the confidence of the comparison results. And also the so called "cell-cell interaction" is purely based on potential ligand-receptor pairs

expressed by non-RGCs and RGCs separately, more experimental results are needed to prove that is the case in vivo. Although a lot of data analysis with many pathways presented, there is no real new knowledge comes out from this study. It would be helpful if the novelty and new findings can be presented in a more clearcut way.

Recommendations for the authors

Some aspects of paper writing, image presentation and data analysis need improvement.

- 1) Fig. 1b: What is the numbers of each individual retinal cell types? Without positive selection, rare cell populations are notoriously difficult to acquire, for example microglia and RGCs. It is heavily biased towards larger populations. It would be more informative to show the absolute numbers and percentage of each types of retinal cells, before and after crush, and any difference (whether RGC axon injury induces increase or decrease of certain glial cell types?). Are there enough cells to have meaningful data analysis? On line 125, it seem 1055 RGCs are collected, however, RGCs are only 1% of retinal cells and even without any lost (assuming 100% recovery), totally collected retinal cells are supposedly to 105,500 cells. However, they only reported 56,531 total retinal cells collected. How that is possible?
- 2) Fig. 1b: Are there heterogeneity among muller glia? How about RodBC? Cones? HCs? The UMAP shows sub-structure in them, are they due to batch effect? Time points after injury? Quality issue? Or actual subtypes among types? The activated Muller glia shares some gene features with astrocyte. How are the two separated? Benhar et al. 2023 paper described in more details the gene changes from subtypes of muller glia/astrocyte/micro glia/ immune populations upon injury. It would be interesting to know if the subtypes were identified in the current data, and if the LR pairs showed any differences among them.
- 3) Fig. 1e: line 113-115 "Moreover, genes involved in protein translation and glycolytic metabolism were upregulated in almost all retinal cell types (Fig. 1e, Supplementary Fig. 1)", It's difficult to appreciate where are upregulated protein translation genes in the Fig. 1e and Suppl. Fig. 1, can you clarify? And it is well known RGC protein translation is shut off after optic nerve crush injury, better to show RGC data as well.
- 4) Fig. 1g: which groups are compared to generate the fold change, 12hr vs sham, 1d vs sham, or 2d vs sham? Or it is pooled together? What we can learn from the different time points? Are the ligand-receptor pairs are present in non-RGCs and RGCs separately to be considered as cell-cell interaction effect or did not differentiate at all?
- 5) Suppl. Fig. 2a: the correlation is done with naïve RGCs or pooled naïve and injured RGCs? It needs more evidence showing the Tran et al 2019 dataset is a valid surrogate for this study. Not even sure this can be done this way. Suppl. Fig. 2a showed that many genes have different levels of expressions between the two datasets. More importantly, it's not clear if the injury induced changes are consistent, and therefore raise the question whether replacing the RGCs with those from another dataset is valid for the ligand-receptor analysis. More through comparison is needed. In principle, should be done within the same dataset from the same preparation.
- 6) Fig. 2: overly simplified correlation analysis without considering potential non-RGC cell-cell signaling pairs and RGC autocrine effects.
- 7) Fig2a. Does the number of cells in each class affect the ability to detect LR pairs? Any correlation between cell number and number of LR pairs?
- 8) Fig2a. Which time point of the data was used to generate the figure? How do the number of LR pairs change along the time of injury?
- 9) Fig2a. It's surprising that many of LR pairs exist between photoreceptors and RGCs, or RPEs and RGCs. Are these real interactions or are these computational noises?
- 10) Fig2a/3a. Are the numbers normalized by the number of genes detected per cell?
- 11) Fig. 3d: "Validation of Anxa1-Dysf interaction, from MG to αRGC." Not sure how this figure validates interaction of Anxa2 and Dysf since these signals are from different cell types and layers and not even close to each other? Is Anxa2 secreted? It is a membrane protein, not secreted protein, right?
- 12) Fig3d bottom row, many Dysf+ cells are Spp1-, especially 2d post injury. Is this an alphaRGC-

Muller interaction or a 2dpcRGC-Muller interaction or an otherRGC-Muller interaction?

13) Fig. 7b: normally microbeads model cannot sustain IOP elevation more than 3 weeks and often require 2nd injection, how this is achieved?

14) Fig. 7c: how about axon protection in optic nerve? If there is only RGC soma protection, it is hard to reconcile the visual function effect.

15) Need to make statements more accurate.

For example, regarding line 70-71, "the optic nerve crush (ONC) model, which exhibits similarities in gene expression patterns with glaucoma¹⁷", is certainly wrong. Crush model is an acute axotomy injury that is drastically different to ocular hypertension glaucoma and more so for normal tension glaucoma.

In line 109, the result suggests that the downregulation of genes associated with chromatin assembly and remodeling is transient, however, Muller glia can't spontaneously reprogramming after retinal injury in the mammalian, even when the expression levels of relevant genes bounce back 2d after injury.

Throughout the manuscript, ligand-receptor interaction is used but without experimental proof that the interaction is really happening, it is just co-expression and potential pairs.

Reviewer #3 (Remarks to the Author):

This is a well-designed study which used a combination of bioinformatics tools, in vivo models of retinal injury and mouse behavioral assays to reveal a new role of Oprm1 in promoting the survival of RGCs.

The authors performed droplet-based scRNA-seq on the retina at three time points after the optic nerve crush (ONC) injury. The authors performed LRLoop, a recently developed computational method for analyzing ligand-receptor based cell-cell communication, to explore interactions among RGCs and the retinal cell types. Importantly, they identified the μ -opioid receptor, encoded by the Oprm1 gene, as a novel neuroprotective factor for pan-RGCs. While the roles of μ -opioid receptor in regulating pain, reward, and addictive behaviors are well characterized, its role in promoting RGC survival was not describe previously. The authors elegantly used AAVs and Cre mice to determine if Oprm1 is a neuroprotective factor for RGCs. Strikingly, expression of Oprm1 promoted survival of RGCs and prevented vision function loss in a clinically relevant model of glaucoma. The snRNAseq and the cell-cell interaction dataset provided in this study will be a valuable resource, not only for the investigators in the vision research field, but also for those who study neuroprotective strategies, neurodegeneration and neuronal death mechanisms in general. The data are presented with a high degree of clarity. There are a few issues which the authors could address to further strengthen the study.

Tissue validation of Oprm1 expression (at the mRNA and protein levels) in ipRGCs is lacking. Studies have shown that most ipRGC subtypes (e.g. M1-M6 ipRGCs) have high survival capacity. Tran et al., scRNAseq dataset seem to show that M2 ipRGCs express high level of Oprm1. However, it is unclear if other ipRGC subtypes also express Oprm1. Perhaps the authors could use their own dataset or those from others to clarify the extent to Oprm1 expression in the different ipRGC subtypes.

It is unclear whether or not the AAV2-FLEX-sun1GFP and AAV2-FLEX-hOprm1 induce expression of transgene equally and proportionally in all RGC types. The authors need to use IHC (e.g. antibodies against RGC subtype markers) and characterize the extent of GFP expression in different RGC types.

It is unclear how specific the Oprm1 ligands, proenkephalin (Penk), prodynorphin (Pdyn), and proopiomelanocortin (Pomc) are in activating Oprm1 specifically. It is unclear if these ligands have

other receptors which they can bind to. If there are other receptors which exist and if they are indeed expressed in in retinal cells other than RGCs, then it is plausible that these ligands might act also on different retinal cells to induce RGC protection. In these regards, the mechanisms by which Oprm1 protects RGCs is nuclear. Perhaps the authors could elaborate and discuss these aspects.

Given the recent studies which performed scRNAseq on human retinas, it will be useful if the authors can comment on whether or not Oprm1 is expressed in human RGCs and whether its ligands are expressed in RGCs or in non-RGC cells in the human retina.

It remains a questions whether Oprm1 works in RGCs strictly or the similar mechanism operates in other CNS regions. While Oprm1 is expressed predominantly in ipRGCs in the retina, the pan-RGC protection in Oprm1 overexpressed animals indicate that the machinery downstream of Oprm1 is present in all neurons. The authors could elaborate and provide insights into expression level of Oprm1 and correlation of its natural expression to survival capacity of CNS neurons outside the retina.

Several studies highly relevant to the topics discussed in this article could be cited and added to the reference list including, "Subtype-specific survival and regeneration of retinal ganglion cells in response to injury". PMID: 36035999.

Other minor issues are as follows;

Lines 140, 166, typos are found (i.e. ligand-recpetor).

Scale bars in some images are missing.

Reviewer #1 (Remarks to the Author):

The current study by Qian J. and Zhou F. systematically surveyed cell-cell interactions among neurons and non-neuronal cells in the retina. The work was put in a degeneration (ONC and glaucoma) setting, and the authors provided a systematic survey using their own algorithm. The bioinformatic prediction landed on an interesting hit named Oprm1, enriched in ipRGCs, but has a broad impact on neuroprotection in both injuries and degeneration models.

Overall, the work is innovative, thorough, and interdisciplinary. It provides a model for others in retina neurobiology and neurodegeneration to study biological processes in the cell-cell interaction. This current study conceptually advances the field, rather than the past, focusing on RGC-intrinsic properties. The work is timely, given the ever-exploding scRNA-seq data – the authors combined multidisciplinary approaches and executed a beautiful storyline from in silico prediction to in vivo translational research. The team carried out multiple collaborative experiments at the Hopkins Med school, which was truly impressive as a dream team. Thus, the study would greatly interest general readers in neuroscience, ophthalmology, neurology, and regenerative medicine. The application of innovation of the bioinformatics approaches was also a huge plus for many others to follow.

While the authors focused on Oprm1 biology, I view this paper more like a method/resource for the field to share, as they also highly cited Tran 2019, Neuron. Thus, my major points were listed below but largely on the organization of the existing data, the transparency of the report, as well as the upcoming data sharing/database look-up for general readers to learn and utilize in the field.

RESPONSE: We thank the reviewer for the positive comments.

Major Points:

1) The biggest highlight of this paper is a systematic survey of all known ligand-receptor interactions: (a) for clarity for all readers, especially those outside the bioinformatics field, the authors may want to consider organizing the Ligand-Receptor (L to R) and keep it highly consistent with the figures/text, and also make it very clear to the readers – I am often lost by reading this; (b) a central hypothesis here is that RGCs provide the ligands but the other cells serve as receptors. – could the authors elaborate the data analysis in a reverse manner regarding the potential role of ligands from non-retinal neurons onto RGCs (receptors)? It might be beyond the scope of the current paper, but it would be a complementary resource for the field to understand this intriguing question and emerging field. (c) Are more time points offered in the current study – does that change the interactome? Esp. the Tran paper offered multiple time-points – such as 3-4dpc? 1wpc?

RESPONSE: We appreciate the reviewer's suggestions. (a) We have improved the consistency of the ligand-receptor representation in both the figures and the text. Throughout the manuscript, we use the notation L-R to describe interactions between ligands and receptors, with L representing a ligand and R representing a receptor. (b) The reviewer

correctly pointed out that our analysis involved calculating interactions from retinal ganglion cells (RGCs) to other retinal cells, which we have termed "distress signals." In addition, we also performed an analysis of interactions from other retinal cells to RGCs, which we refer to as "responsive interactions." Importantly, the neuroprotective interactions identified in our study originate from other retinal cells and are directed towards RGCs. (c) Since our study focus on early events following injury before neuronal cell death, our gene expression measurements extend up to 2 days post-injury. If we were to extend our observation beyond 2 days following injury, the majority of RGCs would likely undergo cell death, making it increasingly difficult to interpret cell-cell interactions.

These adjustments have been incorporated into the revised manuscript to improve the clarity.

2) The set of images within Fig 3d provided a solid experimental basis for the bioinformatics prediction – however, the current set of images was very hard to read and interpret as four-color images – The authors should consider providing high-resolution and split 4-channel images in a dedicated supplementary figure.

RESPONSE: We thank the reviewer for the suggestion and provided the single channel images in Supplementary Fig. 3c.

3) Interpretation for Fig 4e raises some questions – the authors need to consider the distance, especially as all RGCs are packed together, providing immediate access to the neighbors – thus, even though the ligand expression level is low, the paracrine effects within RGC groups can be prominent for neuroprotection – these are interesting and unexplored areas which the authors provided the first insights- yet the conclusion and discussion on these points need to be carefully flashed out.

RESPONSE: We agree with the reviewer that our conclusion exclusively focused on cell numbers while overlooking the spatial relationships between cells. In response, we have revised the interpretation of our results accordingly.

4) The transition from Fig 5 to the top of Fig 6 needs more experimental data to verify the findings = Using the same preparations as the authors did in Fig 5l/m with only anti-opn4 staining, the authors should consider integrating more antibody staining (rather than only focusing on Opn4./ipRGCs) – this is a critical piece of data validating the finding. The antibodies should be included to verify the findings include goat anti-Osteopontin/Spp1 R&D Systems Cat#AF808; rabbit anti-Melanopsin ATsbio Cat#AB-N38; rabbit anti-Foxp2Abcam Cat#ab16046; rabbit anti-Tbr1 Cell Signaling Technology Cat#49661; rabbit anti-Satb1 Abcam Cat#ab109122;

RESPONSE: To address the reviewer's concerns, we performed antibody staining with various RGC subclass markers in Oprm1 mCherry tagged mice (see Figures below). Our results clearly demonstrated that Oprm1 predominately expressed in ipRGC (Opn4+) and alpha-RGC (Spp1+) (new Fig. 6c and d, and Suppl. Figs. 7 and 8). On the other hand, the less resilient RGC subclasses exhibit a significantly lower proportion of cells expressing Oprm1. In fact, only 1.46% T-RGC (Tbr1+), 1.43% DS-RGC (Satb1+), and 1.60% F-RGC

(Foxp2+) show Oprm1 expression. We included the new results in Fig. 6c and Supplementary Fig. 7.

5) *The entire Figure 5 is very critical for the manuscript. However, I am confused about the random data points that the authors chose to analyze the end-points – Fig 5b included 14dpi; Figure 5f included 7dpi; Figure I was 5dpi; Figure L was 4dpi – I understand that they were different models – but some systematic organization and comparison, and explanations to the readers/reviewers are highly desired here.*

RESPONSE: As indicated by several previous studies [PMID: 31784286, PMID: 34297923], the percentage of surviving pan-RGCs was approximately 50% at 5 days after optic nerve crush (ONC). By the 14th day after ONC, this percentage reached a plateau, stabilizing at 15-20%, with minimal further decrease even up to 21 or 28 days post-ONC. Consequently, our analysis focused on two critical time points: 5 days and 14 days post-ONC, allowing for the assessment and quantification of the dynamics in RGC cell death. For further investigations into the neuroprotective effects of Oprm1 using morphine in ONC model, we chose a single time point—5 days post-ONC.

Given the quick RGC death induced by NMDA treatment, we adopted the widely accepted standard protocol of a single time point, specifically 7 days post-NMDA treatment, as utilized in other studies [PMID: 34297923]. This time frame allowed us to elucidate the impact of Oprm1 OE on late-stage NMDA toxicity. To explore the influence of naloxone, an Oprm1 antagonist, on ipRGC survival following NMDA excitotoxicity, we selected an early time point—4 days after NMDA injury. At this timepoint, some ipRGCs had died from NMDA toxicity, yet sufficient numbers remained for an examination of the potential effects of loss of function with naloxone. We provided some explanations about the rationale of timepoint selection in the revised manuscript.

6) *Related to Major Point 4 – it would be critical for the authors to carry out double-color immunostaining for Oprm1 [one antibody was applied in Fig 7b, but I am not too sure about the specificity] onto Spp1, Opn4, Tbr1, foxp2, etc? (alternatively, double-color in situ/RNA-scope for Oprm1 onto several markers) – The expression level/percentages must be quantified to further support the findings in both Fig 5/Fig 6a-c.*

RESPONSE: Several lines of evidence suggest that Oprm1 is predominantly expressed in ipRGCs, and the protective effect may extend to other RGC subclasses. First, both Tran's data and our snRNA-seq data demonstrate high Oprm1 expression in ipRGCs. Second, our new antibody staining results validate the Oprm1 expression patterns in various RGC subclasses (refer to Point #4). Third, our snRNA-seq data indicate that the overexpression of Oprm1 in the ONC model protects all RGC subclasses (Figure 6e). Fourth, our new antibody staining results also confirm the overexpression of AAV-induced hOprm1 in multiple RGC subclasses (see Figure below in the response to Reviewer #3). Double-color immunostaining is impractical for the injury models in Fig 5 due to (1) the extended time for experiments, (2) the inaccurate quantification with antibody staining, and (3) the limited additional information it would provide beyond the four lines of evidence.

7) *Related to Fig S7 and the paper in general: The method section did NOT include the antibody information (Catalog, species) for immunohistochemistry and their dilutions unless the information was included somewhere else in the report summary that I neglected.*

RESPONSE: We added the antibody information in the Methods section.

8) *Related to Major Point 1 - In the revised format, the authors would need to lay out their plan to share the big data interactively to lay audience and non-bioinformatics experts – some samples from Tran et al., 2019 at the Broad Institute would be a good example. https://singlecell.broadinstitute.org/single_cell/study/SCP509/mouse-retinal-ganglion-cell-adult-atlas-and-optic-nerve-crush-time-series/
The authors are leading experts in bioinformatics and, thus, should have the bandwidth to share such data with the public in compliance with the NIH Data sharing rules.*

RESPONSE: Thank the reviewer for the suggestion. We uploaded our data to Single Cell Portal hosted by Broad Institute. The users can explore and visualize our data through the website. https://singlecell.broadinstitute.org/single_cell/study/SCP2423/

MINOR Points

1) *Discussion: The study performed their analysis at 2 weeks following optic nerve crush, at which 50-80% RGCs are already lost. This limits the interpretation of the cell-cell interaction data, for example. The statement that "all neighbors are good neighbors" and that neighboring RGCs are helping to promote RGC survival "rather than sending any additional apoptotic signals to destroy injured RGCs" is somewhat misleading- this could be due to the study focusing only on the few surviving cells after 2 weeks. Two weeks could be too long to see negative interactions. If data are available, maybe consider additional time points (days and 1-week post crush) to see how cell-cell interactions change as RGCs are lost would be extremely informative.*

RESPONSE: We collected the samples within the first two days following optic nerve crush, rather than two weeks after the injury. During this initial 2-day period, there is minimal RGC death, allowing us to investigate all cell-cell interactions. A significant portion of RGCs (~80 to 90%) do not survive at two weeks after injury, and thus, we cannot collect the data from these cell death susceptible subtypes. We added more discussions in "all neighbors are good neighbors".

2) *PMID: 22661469 should be considered a citation relevant to the concept here.*

RESPONSE: We included the reference in the revised version.

1) *Line 175 citations - the citation of Ref 32 should be added to 4&27; not sure about Ref 28 as it was a zebrafish paper - unless the authors utilized its data – I would delete it.*

RESPONSE: We fixed it.

2) *Line 221 citations are missing Ref 27 and PMID: 37624696.*

RESPONSE: We fixed it.

5) The readers would benefit from an illustrative model describing the biology of *Oprm1* in terms of the RGCs' neuroprotection.

RESPONSE: As suggested by the reviewer, we made a schematic figure as Supplementary Fig. 10b.

Reviewer #2 (Remarks to the Author):

Summary

Using scRNA-seq with mouse retinal cells before and after optic nerve crush, this group focuses on the potential interaction of glial cells with RGCs mediated by cell surface ligands and receptors. Bioinformatic analysis of transcriptomes of non-RGC retinal cells collected by this group and previously published RGCs, they identified a group of ligand-receptor pairs seems upregulated in surviving RGCs after axotomy. They continue to demonstrate that one of the receptors, *Oprm1*, showed neuroprotection in three optic neuropathy models. Although the finding is interesting and intriguing, the data analysis is overly simplified, especially lacking RGCs from the same batch of non-RGC collections jeopardize the confidence of the comparison results. And also the so called “cell-cell interaction” is purely based on potential ligand-receptor pairs expressed by non-RGCs and RGCs separately, more experimental results are needed to prove that is the case in vivo. Although a lot of data analysis with many pathways presented, there is no real new knowledge comes out from this study. It would be helpful if the novelty and new findings can be presented in a more clearcut way.

RESPONSE: We thank the reviewer for the positive comments (“*the finding is interesting and intriguing*”). However, the reviewer also raised some significant concerns, which we will address in the following discussion.

A. “lacking RGCs from the same batch of non-RGC collections”,

First, it's technically and economically challenging to sequence both RGCs and other non-RGC cells in the same batch due to the fact that RGCs make up only ~1% of retinal cell population. In fact, the reviewer also acknowledged “*rare cell populations are notoriously difficult to acquire, for example microglia and RGCs. It is heavily biased towards larger populations*”. To obtain a comparable number of RGCs as in Tran's work, it would

necessitate an 86-fold increase in our library size. This would result in an exceedingly costly project, with a significant portion of the budget allocated to sequencing abundant retinal cells. Second, if we acknowledge the need to sequence RGCs and non-RGCs in separate batches, the subsequent question is: can we integrate our data with existing published data, or are we required to generate our own RGC sequencing data? We strongly believe that the entire research community should promote data sharing, fostering the discovery of novel biological insights through data integration. As an illustration, Tran's RGC data were also successfully integrated into Benhar's work to identify ligand-receptor interactions (*Nature Immuno.* 2023). Third, we have strong justification to integrate our data with Tran's dataset. We followed the same experimental protocols and conducted our experiments at the same time points. More importantly, the gene expression profiles of the 1,055 RGC cells from our dataset and the 86,426 RGCs collected by Tran displayed a significant level of correlation ($cor = 0.84$), indicating compatibility between the two datasets. To offer a point of reference for the level of correlation, we also calculated the correlation coefficients between RGCs from Benhar's and Tran's studies, where their correlation coefficient reached 0.85, a value comparable to our obtained correlation coefficient (see figure below, note that $2.2e-16$ is the limit of p-value in our calculation).

Fourth, we also assessed the transferability of RGCs data at the interaction level. Taking the interactions from Müller glia (MG) to RGCs as an example, we computed the ligand-receptor interaction strengths using both our RGC dataset and Tran's RGC dataset. Notably, we observed a strong correlation in interaction strengths between the two datasets (see below), indicating our ability to detect similar interactions using both datasets.

Finally, through the integrative analysis, we successfully recovered many established neuroprotective factors and experimentally validated one novel factor, Oprm1. This outcome underscores the success of our data integration efforts.

We added the new results in Supplementary Fig. 2a to support the integration of datasets.

B. *“so called “cell-cell interaction” is purely based on potential ligand-receptor pairs expressed by non-RGCs and RGCs separately, more experimental results are needed to prove that is the case in vivo”*

It is important to acknowledge that it is difficult to validate the ligand-receptor interactions in an in vivo setting. The majority of previously published studies either did not validate their prediction of interactions at all (e.g., Benhar et al, 2023) or validated the predicted interactions by confirming the expression of ligand and receptor genes. As a matter of fact, a significant challenge in the computational prediction of cell-cell interactions lies in the lack of a gold standard dataset for benchmarking the predictions. This challenge stems from the limited availability of experimentally validated cell-cell interactions in the literature, highlighting the difficulties in experimentally examining such interactions in vivo.

In our paper, we validated one of the predicted interactions, Anxa1-Dysf, using RNAscope. Furthermore, we performed in-depth experimental validation of one neuroprotective factor, Oprm1, which was resulted from our predicted ligand-receptor interactions. In fact, a substantial portion of our paper, comprising 3 out of a total of 7 figures, is devoted to the functional validation of the newly identified neuroprotective factor, Oprm1. This validation was conducted in three animal models, including optic nerve crush, NMDA, and elevated intraocular pressure models.

C. *“there is no real new knowledge comes out from this study”*

We respectively disagree with the reviewer. In this paper, we made several new discoveries. Here we highlight a few.

1. We discovered a novel neuroprotective factor, Oprm1, which is known to play roles in regulating pain, reward, and addictive behavior, but has not been previously implicated in regulating neuronal survival.

2. Besides Oprm1, we also predicted several new neuroprotective interactions for RGCs.

3. We also discovered a few general rules regarding the role of cell-cell interactions in neuronal systems. For example, high-survival cells tend to have more interactions with neighboring cells, suggesting that high-survival RGCs are intrinsically programmed to foster more communication with their surroundings.

Recommendations for the authors

Some aspects of paper writing, image presentation and data analysis need improvement.

1) Fig. 1b: What is the numbers of each individual retinal cell types? Without positive selection, rare cell populations are notoriously difficult to acquire, for example microglia and RGCs. It is heavily biased towards larger populations. It would be more informative to show

the absolute numbers and percentage of each types of retinal cells, before and after crush, and any difference (whether RGC axon injury induces increase or decrease of certain glial cell types?). Are there enough cells to have meaningful data analysis? On line 125, it seem 1055 RGCs are collected, however, RGCs are only 1% of retinal cells and even without any lost (assuming 100% recovery), totally collected retinal cells are supposedly to 105,500 cells. However, they only reported 56,531 total retinal cells collected. How that is possible?

RESPONSE: Thank the reviewer for the suggestion. We provide the numbers of individual retinal cell types before and after the injury (see Suppl. Table 1). When estimating the expected total number of retinal cells based on RGC counts, it is important to account for variations in the resilience of different retinal cell types to the dissociation and single-cell sequencing procedure. This variance arises from the structural fragility of certain cell types (e.g., photoreceptors), making them more susceptible to enzymatic and mechanical dissociation methods. Moreover, the removal of low-quality cells during computational analysis may disproportionately affect retinal cell types. Therefore, retinal cell types exhibit varying survival rates during the procedure, which can lead to inaccuracies when attempting to directly estimate the total cell count based solely on known population percentages of individual cell types. Due to the same reason, we are reluctant to over-interpret the number of individual retinal cell types before and after the injury.

2) Fig. 1b: Are there heterogeneity among muller glia? How about RodBC? Cones? HCs? The UMAP shows sub-structure in them, are they due to batch effect? Time points after injury? Quality issue? Or actual subtypes among types? The activated Muller glia shares some gene features with astrocyte. How are the two separated? Benhar et al. 2023 paper described in more details the gene changes from subtypes of muller glia/astrocyte/micro glia/ immune populations upon injury. It would be interesting to know if the subtypes were identified in the current data, and if the LR pairs showed any differences among them.

RESPONSE: Frist, we do observe the heterogeneity within certain clusters. For example, the cone bipolar cells (ConeBC) cluster can be subdivided into ON-ConeBC and OFF-ConeBC (see Figure below).

Moreover, the heterogeneity is not attributed to batch effects, as the UMAPs remain consistent across different time points (see Figure below).

In our data, Muller glia and astrocytes can be separated by their respective marker genes (see Figure 1d).

Second, it is important to emphasize that our intention is not to create a comprehensive retinal atlas by generating and characterizing the subtypes for each retinal cell types. Benhar et al. have already done a great job in identifying subtypes of the major retinal cell types and the temporal expression patterns of these subtypes. We do not want to replicate the effort in this work. Instead, the focus of our work is to identify cell-cell interactions between RGCs and other retinal cell types. We demonstrate the temporal patterns of LR pairs in Figure 2C and Fig. S2E. To identify the neuroprotective interactions, we have already generated ligand-receptor interaction data between 9 RGC subclasses and 13 retinal cell types, resulting in a substantial dataset comprising 117 pairs of cell types. If we were to further dissect the cell-cell interactions within each sub-cluster of every retinal cell type, the project would quickly become unmanageable. Lastly, we acknowledge that there may be important ligand-receptor interactions between subtypes of RGCs and subtypes of each retinal cell type. This is an avenue we plan to explore in our future projects, and any resulting findings will be subject to additional publications.

3) Fig. 1e: line 113-115 “Moreover, genes involved in protein translation and glycolytic metabolism were upregulated in almost all retinal cell types (Fig. 1e, Supplementary Fig. 1)”, It’s difficult to appreciate where are upregulated protein translation genes in the Fig. 1e and Suppl. Fig. 1, can you clarify? And it is well known RGC protein translation is shut off after optic nerve crush injury, better to show RGC data as well.

RESPONSE: We appreciate the reviewer for bringing up the concern. The sentence was intended to refer to Fig. 1f and Suppl. Fig. 1a-c, not Fig. 1e. We corrected the error. To improve the clarity, we have rephrased the sentence as follows: “Moreover, GO analyses have demonstrated a temporal increase in glycolytic activity, ribosomal subunit assembly, cytoplasmic translation, peptide biosynthesis, and oxidative phosphorylation processes within the two-day timeframe following ONC. This increase was observed in retinal ganglion cell (RGC) neighboring cell types, including Müller glial cells (MG), astrocytes, microglia, or amacrine cells (Fig. 1f and Suppl. Fig. 1a-c)... “

4) Fig. 1g: which groups are compared to generate the fold change, 12hr vs sham, 1d vs sham, or 2d vs sham? Or it is pooled together? What we can learn from the different time points? Are the ligand-receptor pairs are present in non-RGCs and RGCs separately to be considered as cell-cell interaction effect or did not differentiate at all?

RESPONSE: In this analysis, fold change values obtained from comparing 12 hours vs. sham, 1 day vs. sham, and 2 days vs. sham were combined. For each gene, the highest absolute $|\log_2FC|$ value among these three comparison groups was utilized in generating the figures. We made it clear in the Figure legend. Note that in this figure we did not investigate ligand-receptor interactions between RGCs and non-RGCs. Instead, we focused on the transcriptomic responses within non-RGCs and conducted a gene expression analysis separately for the ligand and receptor genes within these non-RGCs, without taking into account their interactions with RGCs. Our analysis of ligand-receptor interactions starts from Figure 2.

5) Suppl. Fig. 2a: the correlation is done with naïve RGCs or pooled naïve and injured RGCs? It needs more evidence showing the Tran et al 2019 dataset is a valid surrogate for this study. Not even sure this can be done this way. Suppl. Fig. 2a showed that many genes have different levels of expressions between the two datasets. More importantly, it's not clear if the injury induced changes are consistent, and therefore raise the question whether replacing the RGCs with those from another dataset is valid for the ligand-receptor analysis. More through comparison is needed. In principle, should be done within the same dataset from the same preparation.

RESPONSE: Please see our response in the first section. The correlation was done with all RGCs combined.

6) Fig. 2: overly simplified correlation analysis without considering potential non-RGC cell-cell signaling pairs and RGC autocrine effects.

RESPONSE: The focus of the paper is the interactions between the RGCs and other retinal cells. As suggested by the reviewer, we performed an analysis to identify the interactions between non-RGC cells. We incorporated the new results in Supplementary Fig. 10a, which included both autocrine and paracrine interactions. In addition, we also analyzed the RGC autocrine effects in Fig. 4e, showing that the protective interactions were achieved through both paracrine or autocrine signaling.

7) Fig2a. Does the number of cells in each class affect the ability to detect LR pairs? Any correlation between cell number and number of LR pairs?

RESPONSE: It is clear that the cell number does not affect the ability to detect LR pairs. For instance, astrocytes have a relatively small population of cells (Fig. 1b), yet we identified the highest number of L-R pairs with RGCs. Conversely, photoreceptors (rods and cones) have a larger cell number (Fig. 1b), but they establish the fewest L-R pairs with RGCs. Overall, there is no correlation between the cell numbers and LR pairs (see below). We included the new result in Supplementary Fig. 2b.

8) Fig2a. Which time point of the data was used to generate the figure? How do the number of LR pairs change along the time of injury?

RESPONSE: We calculated the total unique LR pairs across the four time points and showed the numbers in Fig 2a. The number of LR pairs remains relatively stable throughout the time course (see below). We included the result in Supplementary Fig. 2c.

9) Fig2a. It's surprising that many of LR pairs exist between photoreceptors and RGCs, or RPEs and RGCs. Are these real interactions or are these computational noises?

RESPONSE: The reviewer raised an interesting question. First, we want to highlight that the number of LR pairs detected between photoreceptors and RGCs is significantly lower than those between other cell types and RGCs. Even though our LRLoop algorithm does not incorporate spatial distribution information, the limited number of predicted LR pairs between photoreceptors and RGCs underscores the efficacy of the algorithms. Second, no prior studies have definitively stated the absence of interactions between these spatially separated cell types. Secreted ligands may still be able to migrate between these cell types. Last, our current computational methods cannot distinguish between direct and indirect interactions among cell types. It is possible that some interactions are mediated through intermediary cell types.

10) Fig2a/3a. Are the numbers normalized by the number of genes detected per cell?

RESPONSE: The number of L-R pairs and sum of S_{LR} are not normalized by the number of genes detected per cell.

11) Fig. 3d: "Validation of Anxa1-Dysf interaction, from MG to α RGC." Not sure how this figure validates interaction of Anxa2 and Dysf since these signals are from different cell types and layers and not even close to each other? Is Anxa2 secreted? It is a membrane protein, not secreted protein, right?

RESPONSE: First, it is well-established that Muller glia interact with neurons such as RGCs to regulate various biological processes (see a review "Glia-neuron interactions in the mammalian retina" PMID: 26113209). A schematic figure from this review illustrates the relative positioning of Muller glia in relation to other retinal cells (see below). Although the cell bodies of Müller glia reside in the inner nuclear layer (INL), their endfeet extend into the ganglion cell layer (GCL), where the somas of RGCs are located. Second, according to previous studies, Anxa1 can undergo externalized and/or secreted, functioning in an autocrine, paracrine and juxtacrine (involving cell-cell contact) manner to activate the downstream signaling (PMID: 19104500).

12) Fig3d bottom row, many *Dysf*⁺ cells are *Spp1*⁻, especially 2d post injury. Is this an alphaRGC-Muller interaction or a 2dpcRGC-Muller interaction or an otherRGC-Muller interaction?

RESPONSE: According to our scRNA-seq data, we identified *Anxa1*-*Dysf* as a ligand-receptor pair between Müller glia and α RGC. In Figure 3d, many non- α RGC cells also express *Dysf*. However, the expression of *Dysf* is most pronounced in α RGC, exhibiting significant changes following injury (see figure below). We made it clear in the revised manuscript.

13) Fig. 7b: normally microbeads model cannot sustain IOP elevation more than 3 weeks and often require 2nd injection, how this is achieved?

RESPONSE: We adopted a method from a previous published paper (see below for Figure 2 in PMID: 27077732), in which the IOP can sustain for at least six weeks. Briefly, we used magnetic microbeads for intracameral injection and used magnets to draw the microbeads towards the circumference between the cornea and iris. With this method, we successfully maintained elevated IOP for up to seven to eight weeks.

14) Fig. 7c: how about axon protection in optic nerve? If there is only RGC soma protection, it is hard to reconcile the visual function effect.

RESPONSE: We thank the reviewer for the insightful question. Our vision-guided behavior experiments have revealed that the overexpression of Oprm1 enhances vision in the glaucoma model, suggesting the potential protective effect on optic nerve. To validate the protective effect on optic nerve, we utilized toluidine blue staining to visualize myelinated nerve fibers. The figure illustrates cross-sections of the optic nerve, located 1 mm from the eyeball. The overexpression of Oprm1 resulted in a substantial improvement in axon survival in eIOP model, increasing from 66.5% to 84.3%. This finding aligns with the observed survival rate of retinal ganglion cell somas. We include the result in Fig. 7e and f.

15) Need to make statements more accurate.

For example, regarding line 70-71, “the optic nerve crush (ONC) model, which exhibits similarities in gene expression patterns with glaucoma17”, is certainly wrong. Crush model is an acute axonomy injury that is drastically different to ocular hypertension glaucoma and more so for normal tension glaucoma.

RESPONSE: Following the reviewer’s suggestion, we deleted the sentence.

In line 109, the result suggests that the downregulation of genes associated with chromatin assembly and remodeling is transit, however, Muller glia can’t spontaneously reprogramming after retinal injury in the mammalian, even when the expression levels of relevant genes bounce back 2d after injury.

RESPONSE: These are the genes enriched in our Gene Ontology analysis. We did not intend to imply that Müller glia spontaneously undergo reprogramming after retinal injury following the downregulation of these genes.

Throughout the manuscript, ligand-receptor interaction is used but without experimental proof that the interaction is really happening, it is just co-expression and potential pairs.

RESPONSE: Please see our response in the first section.

Reviewer #3 (Remarks to the Author):

This is a well-designed study which used a combination of bioinformatics tools, in vivo models of retinal injury and mouse behavioral assays to reveal a new role of Oprm1 in promoting the survival of RGCs.

The authors performed droplet-based scRNA-seq on the retina at three time points after the optic nerve crush (ONC) injury. The authors performed LRLoop, a recently developed computational method for analyzing ligand-receptor based cell-cell communication, to explore interactions among RGCs and the retinal cell types. Importantly, they identified the u-opioid receptor, encoded by the Oprm1 gene, as a novel neuroprotective factor for pan-RGCs. While the roles of u-opioid receptor in regulating pain, reward, and addictive behaviors are well characterized, its role in promoting RGC survival was not describe previously. The authors elegantly used AAVs and Cre mice to determine if Oprm1 is a neuroprotective factor for RGCs. Strikingly, expression of Oprm1 promoted survival of RGCs and prevented vision function loss in a clinically relevant model of glaucoma. The snRNAseq and the cell-cell interaction dataset provided in this study will be a valuable resource, not only for the investigators in the vision research field, but also for those who study neuroprotective strategies, neurodegeneration and neuronal death mechanisms in general. The data are presented with a high degree of clarity. There are a few issues which the authors could address to further strengthen the study.

RESPONSE: We thank the reviewer for the positive comments.

Tissue validation of Oprm1 expression (at the mRNA and protein levels) in ipRGCs is lacking. Studies have shown that most ipRGC subtypes (e.g. M1-M6 ipRGCs) have high survival capacity. Tran et al., scRNAseq dataset seem to show that M2 ipRGCs express high level of Oprm1. However, it is unclear if other ipRGC subtypes also express Oprm1. Perhaps the authors could use their own dataset or those from others to clarify the extent to Oprm1 expression in the different ipRGC subtypes.

RESPONSE: To address the concerns raised by the reviewer, we initially analyzed Oprm1 expression patterns across ipRGC subtypes (M1-M6) utilizing Tran's data. Our snRNA-seq using sorted RGC nuclei also indicated a preferential high expression of Oprm1 in ipRGCs M2 and M4 (also known as the alpha-ON-S) subtypes. Subsequently, our own snRNA-seq data validated these results (see figure below).

As M2 and M4 are primarily characterized by *Opn4* and *Spp1* marker genes, we conducted antibody staining in *Oprm1* mCherry-tagged mice (See Figure below). The results revealed that ~80% of the *Oprm1*-expressing cells are either *Opn4*⁺ or *Spp1*⁺. This data is included as the new Fig. 6c and d, and Supplementary Fig. 10c-e.

It is unclear whether or not the AAV2-FLEX-sun1GFP and AAV2-FLEX-hOprm1 induce expression of transgene equally and proportionally in all RGC types. The authors need to use IHC (e.g. antibodies against RGC subtype markers) and characterize the extent of GFP expression in different RGC types.

RESPONSE: We agree with the reviewer that AAV2-FLEX-sun1GFP may not equally and proportionally induce transgene expression in all RGC types. We used it as a control virus, as existing studies did not suggest any neuroprotective effects associated with the overexpression of Sun1GFP.

As suggested by the reviewer, we performed the antibody staining with various RGC subclass marker. Our results shows that the AAV2-FLEX-hOprm1-mCherry induced high Oprm1 expression coverage of pan-RGC, and covering multiple RGC subclasses we examined (see Figure below). We include the result in Supplementary Fig. 8.

It is unclear how specific the Oprm1 ligands, proenkephalin (Penk), prodynorphin (Pdyn), and proopiomelanocortin (Pomc) are in activating Oprm1 specifically. It is unclear if these ligands have other receptors which they can bind to. If there are other receptors which exist and if they are indeed expressed in retinal cells other than RGCs, then it is plausible that these ligands might act also on different retinal cells to induce RGC protection. In these regards, the mechanisms by which Oprm1 protects RGCs is unclear. Perhaps the authors could elaborate and discuss these aspects.

RESPONSE: We have identified additional receptors such as Oprd1, Oprk1, Mrgprx1, and Mc1r, that interact with Penk, Pdyn, and Pomc. We cannot rule out the possibility that expression of these receptors in other non-RGC cells could be neuroprotective. Nevertheless, expression of these receptors in RGCs are unlikely to be neuroprotective based on our prediction, as they do not show significant differential interactions between high- and low-survival RGC subclasses. Note that in our study we directly manipulated Oprm1 in RGCs and showed that it had neuroprotective effects.

Given the recent studies which performed scRNAseq on human retinas, it will be useful if the authors can comment on whether or not Oprm1 is expressed in human RGCs and whether its ligands are expressed in RGCs or in non-RGC cells in the human retina. It remains a question whether Oprm1 works in RGCs strictly or the similar mechanism operates in other CNS regions. While Oprm1 is expressed predominantly in ipRGCs in the retina, the pan-RGC protection in Oprm1 overexpressed animals indicate that the machinery downstream of Oprm1 is present in all neurons. The authors could elaborate and provide insights into expression level of Oprm1 and correlation of its natural expression to survival capacity of CNS neurons outside the retina.

RESPONSE: We have analyzed the gene expression profile of Oprm1 in the human retina [PMID: 32555229]. Interestingly, Oprm1 is predominately expressed in RGC5 cluster, which corresponds to ipRGC subclass. This finding suggests the potential applicability of the protective effect to human RGCs. Whether Oprm1 has protective effects on other CNS neurons would be an interesting topic to pursue in the future.

Several studies highly relevant to the topics discussed in this article could be cited and added to the reference list including, "Subtype-specific survival and regeneration of retinal ganglion cells in response to injury". PMID: 36035999.

RESPONSE: We added the reference.

*Other minor issues are as follows;
Lines 140, 166, typos are found (i.e. ligand-receptor).
Scale bars in some images are missing.*

RESPONSE: We fixed the typos and added the missing scale bars.

REVIEWERS' COMMENTS

Reviewer #1 (Remarks to the Author):

The authors have effectively addressed all of my comments. Their study is both timely and highly innovative. The manuscript, along with the accompanying data, has provided a new platform for understanding neuroprotection specific to RGC types, all while placing the study in a cell-cell interaction setting. As a result, it is well-suited for general readers in the fields of ophthalmology, vision neuroscience, regenerative medicine, and bioinformatics. I recommend this paper for consideration for publication at Nature Communications.

Reviewer #2 (Remarks to the Author):

The revised manuscript has been significantly improved, however, fundamentally the experimental design to determine cell-cell interaction using scRNA-seq is not convincingly sound, and the limited success of identifying single neuroprotective factor with not very striking neuroprotection effect does not boost confidence on the usefulness of this strategy.

Reviewer #3 (Remarks to the Author):

The authors have addressed the concerns raised by this reviewer.

Reviewer #1 (Remarks to the Author):

The current study by Qian J. and Zhou F. systematically surveyed cell-cell interactions among neurons and non-neuronal cells in the retina. The work was put in a degeneration (ONC and glaucoma) setting, and the authors provided a systematic survey using their own algorithm. The bioinformatic prediction landed on an interesting hit named Oprm1, enriched in ipRGCs, but has a broad impact on neuroprotection in both injuries and degeneration models.

Overall, the work is innovative, thorough, and interdisciplinary. It provides a model for others in retina neurobiology and neurodegeneration to study biological processes in the cell-cell interaction. This current study conceptually advances the field, rather than the past, focusing on RGC-intrinsic properties. The work is timely, given the ever-exploding scRNA-seq data – the authors combined multidisciplinary approaches and executed a beautiful storyline from in silico prediction to in vivo translational research. The team carried out multiple collaborative experiments at the Hopkins Med school, which was truly impressive as a dream team. Thus, the study would greatly interest general readers in neuroscience, ophthalmology, neurology, and regenerative medicine. The application of innovation of the bioinformatics approaches was also a huge plus for many others to follow.

While the authors focused on Oprm1 biology, I view this paper more like a method/resource for the field to share, as they also highly cited Tran 2019, Neuron. Thus, my major points were listed below but largely on the organization of the existing data, the transparency of the report, as well as the upcoming data sharing/database look-up for general readers to learn and utilize in the field.

RESPONSE: We thank the reviewer for the positive comments.

Major Points:

1) The biggest highlight of this paper is a systematic survey of all known ligand-receptor interactions: (a) for clarity for all readers, especially those outside the bioinformatics field, the authors may want to consider organizing the Ligand-Receptor (L to R) and keep it highly consistent with the figures/text, and also make it very clear to the readers – I am often lost by reading this; (b) a central hypothesis here is that RGCs provide the ligands but the other cells serve as receptors. – could the authors elaborate the data analysis in a reverse manner regarding the potential role of ligands from non-retinal neurons onto RGCs (receptors)? It might be beyond the scope of the current paper, but it would be a complementary resource for the field to understand this intriguing question and emerging field. (c) Are more time points offered in the current study – does that change the interactome? Esp. the Tran paper offered multiple time-points – such as 3-4dpc? 1wpc?

RESPONSE: We appreciate the reviewer's suggestions. (a) We have improved the consistency of the ligand-receptor representation in both the figures and the text. Throughout the manuscript, we use the notation L-R to describe interactions between ligands and receptors, with L representing a ligand and R representing a receptor. (b) The reviewer

correctly pointed out that our analysis involved calculating interactions from retinal ganglion cells (RGCs) to other retinal cells, which we have termed "distress signals." In addition, we also performed an analysis of interactions from other retinal cells to RGCs, which we refer to as "responsive interactions." Importantly, the neuroprotective interactions identified in our study originate from other retinal cells and are directed towards RGCs. (c) Since our study focus on early events following injury before neuronal cell death, our gene expression measurements extend up to 2 days post-injury. If we were to extend our observation beyond 2 days following injury, the majority of RGCs would likely undergo cell death, making it increasingly difficult to interpret cell-cell interactions.

These adjustments have been incorporated into the revised manuscript to improve the clarity.

2) The set of images within Fig 3d provided a solid experimental basis for the bioinformatics prediction – however, the current set of images was very hard to read and interpret as four-color images – The authors should consider providing high-resolution and split 4-channel images in a dedicated supplementary figure.

RESPONSE: We thank the reviewer for the suggestion and provided the single channel images in Supplementary Fig. 3c.

3) Interpretation for Fig 4e raises some questions – the authors need to consider the distance, especially as all RGCs are packed together, providing immediate access to the neighbors – thus, even though the ligand expression level is low, the paracrine effects within RGC groups can be prominent for neuroprotection – these are interesting and unexplored areas which the authors provided the first insights- yet the conclusion and discussion on these points need to be carefully flashed out.

RESPONSE: We agree with the reviewer that our conclusion exclusively focused on cell numbers while overlooking the spatial relationships between cells. In response, we have revised the interpretation of our results accordingly.

4) The transition from Fig 5 to the top of Fig 6 needs more experimental data to verify the findings = Using the same preparations as the authors did in Fig 5l/m with only anti-opn4 staining, the authors should consider integrating more antibody staining (rather than only focusing on Opn4./ipRGCs) – this is a critical piece of data validating the finding. The antibodies should be included to verify the findings include goat anti-Osteopontin/Spp1 R&D Systems Cat#AF808; rabbit anti-Melanopsin ATSbio Cat#AB-N38; rabbit anti-Foxp2Abcam Cat#ab16046; rabbit anti-Tbr1 Cell Signaling Technology Cat#49661; rabbit anti-Satb1 Abcam Cat#ab109122;

RESPONSE: To address the reviewer's concerns, we performed antibody staining with various RGC subclass markers in Oprm1 mCherry tagged mice (see Figures below). Our results clearly demonstrated that Oprm1 predominately expressed in ipRGC (Opn4+) and alpha-RGC (Spp1+) (new Fig. 6c and d, and Suppl. Figs. 7 and 8). On the other hand, the less resilient RGC subclasses exhibit a significantly lower proportion of cells expressing Oprm1. In fact, only 1.46% T-RGC (Tbr1+), 1.43% DS-RGC (Satb1+), and 1.60% F-RGC

(Foxp2+) show Oprm1 expression. We included the new results in Fig. 6c and Supplementary Fig. 7.

5) *The entire Figure 5 is very critical for the manuscript. However, I am confused about the random data points that the authors chose to analyze the end-points – Fig 5b included 14dpi; Figure 5f included 7dpi; Figure I was 5dpi; Figure L was 4dpi – I understand that they were different models – but some systematic organization and comparison, and explanations to the readers/reviewers are highly desired here.*

RESPONSE: As indicated by several previous studies [PMID: 31784286, PMID: 34297923], the percentage of surviving pan-RGCs was approximately 50% at 5 days after optic nerve crush (ONC). By the 14th day after ONC, this percentage reached a plateau, stabilizing at 15-20%, with minimal further decrease even up to 21 or 28 days post-ONC. Consequently, our analysis focused on two critical time points: 5 days and 14 days post-ONC, allowing for the assessment and quantification of the dynamics in RGC cell death. For further investigations into the neuroprotective effects of Oprm1 using morphine in ONC model, we chose a single time point—5 days post-ONC.

Given the quick RGC death induced by NMDA treatment, we adopted the widely accepted standard protocol of a single time point, specifically 7 days post-NMDA treatment, as utilized in other studies [PMID: 34297923]. This time frame allowed us to elucidate the impact of Oprm1 OE on late-stage NMDA toxicity. To explore the influence of naloxone, an Oprm1 antagonist, on ipRGC survival following NMDA excitotoxicity, we selected an early time point—4 days after NMDA injury. At this timepoint, some ipRGCs had died from NMDA toxicity, yet sufficient numbers remained for an examination of the potential effects of loss of function with naloxone. We provided some explanations about the rationale of timepoint selection in the revised manuscript.

6) *Related to Major Point 4 – it would be critical for the authors to carry out double-color immunostaining for Oprm1 [one antibody was applied in Fig 7b, but I am not too sure about the specificity] onto Spp1, Opn4, Tbr1, foxp2, etc? (alternatively, double-color in situ/RNA-scope for Oprm1 onto several markers) – The expression level/percentages must be quantified to further support the findings in both Fig 5/Fig 6a-c.*

RESPONSE: Several lines of evidence suggest that Oprm1 is predominantly expressed in ipRGCs, and the protective effect may extend to other RGC subclasses. First, both Tran's data and our snRNA-seq data demonstrate high Oprm1 expression in ipRGCs. Second, our new antibody staining results validate the Oprm1 expression patterns in various RGC subclasses (refer to Point #4). Third, our snRNA-seq data indicate that the overexpression of Oprm1 in the ONC model protects all RGC subclasses (Figure 6e). Fourth, our new antibody staining results also confirm the overexpression of AAV-induced hOprm1 in multiple RGC subclasses (see Figure below in the response to Reviewer #3). Double-color immunostaining is impractical for the injury models in Fig 5 due to (1) the extended time for experiments, (2) the inaccurate quantification with antibody staining, and (3) the limited additional information it would provide beyond the four lines of evidence.

7) *Related to Fig S7 and the paper in general: The method section did NOT include the antibody information (Catalog, species) for immunohistochemistry and their dilutions unless the information was included somewhere else in the report summary that I neglected.*

RESPONSE: We added the antibody information in the Methods section.

8) *Related to Major Point 1 - In the revised format, the authors would need to lay out their plan to share the big data interactively to lay audience and non-bioinformatics experts – some samples from Tran et al., 2019 at the Broad Institute would be a good example. https://singlecell.broadinstitute.org/single_cell/study/SCP509/mouse-retinal-ganglion-cell-adult-atlas-and-optic-nerve-crush-time-series/
The authors are leading experts in bioinformatics and, thus, should have the bandwidth to share such data with the public in compliance with the NIH Data sharing rules.*

RESPONSE: Thank the reviewer for the suggestion. We uploaded our data to Single Cell Portal hosted by Broad Institute. The users can explore and visualize our data through the website. https://singlecell.broadinstitute.org/single_cell/study/SCP2423/

MINOR Points

1) *Discussion: The study performed their analysis at 2 weeks following optic nerve crush, at which 50-80% RGCs are already lost. This limits the interpretation of the cell-cell interaction data, for example. The statement that "all neighbors are good neighbors" and that neighboring RGCs are helping to promote RGC survival "rather than sending any additional apoptotic signals to destroy injured RGCs" is somewhat misleading- this could be due to the study focusing only on the few surviving cells after 2 weeks. Two weeks could be too long to see negative interactions. If data are available, maybe consider additional time points (days and 1-week post crush) to see how cell-cell interactions change as RGCs are lost would be extremely informative.*

RESPONSE: We collected the samples within the first two days following optic nerve crush, rather than two weeks after the injury. During this initial 2-day period, there is minimal RGC death, allowing us to investigate all cell-cell interactions. A significant portion of RGCs (~80 to 90%) do not survive at two weeks after injury, and thus, we cannot collect the data from these cell death susceptible subtypes. We added more discussions in “all neighbors are good neighbors”.

2) *PMID: 22661469 should be considered a citation relevant to the concept here.*

RESPONSE: We included the reference in the revised version.

1) *Line 175 citations - the citation of Ref 32 should be added to 4&27; not sure about Ref 28 as it was a zebrafish paper - unless the authors utilized its data – I would delete it.*

RESPONSE: We fixed it.

2) *Line 221 citations are missing Ref 27 and PMID: 37624696.*

RESPONSE: We fixed it.

5) The readers would benefit from an illustrative model describing the biology of *Oprm1* in terms of the RGCs' neuroprotection.

RESPONSE: As suggested by the reviewer, we made a schematic figure as Supplementary Fig. 10b.

Reviewer #2 (Remarks to the Author):

Summary

Using scRNA-seq with mouse retinal cells before and after optic nerve crush, this group focuses on the potential interaction of glial cells with RGCs mediated by cell surface ligands and receptors. Bioinformatic analysis of transcriptomes of non-RGC retinal cells collected by this group and previously published RGCs, they identified a group of ligand-receptor pairs seems upregulated in surviving RGCs after axotomy. They continue to demonstrate that one of the receptors, *Oprm1*, showed neuroprotection in three optic neuropathy models. Although the finding is interesting and intriguing, the data analysis is overly simplified, especially lacking RGCs from the same batch of non-RGC collections jeopardize the confidence of the comparison results. And also the so called “cell-cell interaction” is purely based on potential ligand-receptor pairs expressed by non-RGCs and RGCs separately, more experimental results are needed to prove that is the case in vivo. Although a lot of data analysis with many pathways presented, there is no real new knowledge comes out from this study. It would be helpful if the novelty and new findings can be presented in a more clearcut way.

RESPONSE: We thank the reviewer for the positive comments (“*the finding is interesting and intriguing*”). However, the reviewer also raised some significant concerns, which we will address in the following discussion.

A. “lacking RGCs from the same batch of non-RGC collections”,

First, it's technically and economically challenging to sequence both RGCs and other non-RGC cells in the same batch due to the fact that RGCs make up only ~1% of retinal cell population. In fact, the reviewer also acknowledged “*rare cell populations are notoriously difficult to acquire, for example microglia and RGCs. It is heavily biased towards larger populations*”. To obtain a comparable number of RGCs as in Tran's work, it would

necessitate an 86-fold increase in our library size. This would result in an exceedingly costly project, with a significant portion of the budget allocated to sequencing abundant retinal cells. Second, if we acknowledge the need to sequence RGCs and non-RGCs in separate batches, the subsequent question is: can we integrate our data with existing published data, or are we required to generate our own RGC sequencing data? We strongly believe that the entire research community should promote data sharing, fostering the discovery of novel biological insights through data integration. As an illustration, Tran's RGC data were also successfully integrated into Benhar's work to identify ligand-receptor interactions (*Nature Immuno.* 2023). Third, we have strong justification to integrate our data with Tran's dataset. We followed the same experimental protocols and conducted our experiments at the same time points. More importantly, the gene expression profiles of the 1,055 RGC cells from our dataset and the 86,426 RGCs collected by Tran displayed a significant level of correlation ($cor = 0.84$), indicating compatibility between the two datasets. To offer a point of reference for the level of correlation, we also calculated the correlation coefficients between RGCs from Benhar's and Tran's studies, where their correlation coefficient reached 0.85, a value comparable to our obtained correlation coefficient (see figure below, note that $2.2e-16$ is the limit of p-value in our calculation).

Fourth, we also assessed the transferability of RGCs data at the interaction level. Taking the interactions from Müller glia (MG) to RGCs as an example, we computed the ligand-receptor interaction strengths using both our RGC dataset and Tran's RGC dataset. Notably, we observed a strong correlation in interaction strengths between the two datasets (see below), indicating our ability to detect similar interactions using both datasets.

Finally, through the integrative analysis, we successfully recovered many established neuroprotective factors and experimentally validated one novel factor, Oprm1. This outcome underscores the success of our data integration efforts.

We added the new results in Supplementary Fig. 2a to support the integration of datasets.

B. *“so called “cell-cell interaction” is purely based on potential ligand-receptor pairs expressed by non-RGCs and RGCs separately, more experimental results are needed to prove that is the case in vivo”*

It is important to acknowledge that it is difficult to validate the ligand-receptor interactions in an in vivo setting. The majority of previously published studies either did not validate their prediction of interactions at all (e.g., Benhar et al, 2023) or validated the predicted interactions by confirming the expression of ligand and receptor genes. As a matter of fact, a significant challenge in the computational prediction of cell-cell interactions lies in the lack of a gold standard dataset for benchmarking the predictions. This challenge stems from the limited availability of experimentally validated cell-cell interactions in the literature, highlighting the difficulties in experimentally examining such interactions in vivo. In our paper, we validated one of the predicted interactions, Anxa1-Dysf, using RNAscope. Furthermore, we performed in-depth experimental validation of one neuroprotective factor, Oprm1, which was resulted from our predicted ligand-receptor interactions. In fact, a substantial portion of our paper, comprising 3 out of a total of 7 figures, is devoted to the functional validation of the newly identified neuroprotective factor, Oprm1. This validation was conducted in three animal models, including optic nerve crush, NMDA, and elevated intraocular pressure models.

C. *“there is no real new knowledge comes out from this study”*

We respectively disagree with the reviewer. In this paper, we made several new discoveries. Here we highlight a few.

1. We discovered a novel neuroprotective factor, Oprm1, which is known to play roles in regulating pain, reward, and addictive behavior, but has not been previously implicated in regulating neuronal survival.

2. Besides Oprm1, we also predicted several new neuroprotective interactions for RGCs.

3. We also discovered a few general rules regarding the role of cell-cell interactions in neuronal systems. For example, high-survival cells tend to have more interactions with neighboring cells, suggesting that high-survival RGCs are intrinsically programmed to foster more communication with their surroundings.

Recommendations for the authors

Some aspects of paper writing, image presentation and data analysis need improvement.

1) Fig. 1b: What is the numbers of each individual retinal cell types? Without positive selection, rare cell populations are notoriously difficult to acquire, for example microglia and RGCs. It is heavily biased towards larger populations. It would be more informative to show

the absolute numbers and percentage of each types of retinal cells, before and after crush, and any difference (whether RGC axon injury induces increase or decrease of certain glial cell types?). Are there enough cells to have meaningful data analysis? On line 125, it seem 1055 RGCs are collected, however, RGCs are only 1% of retinal cells and even without any lost (assuming 100% recovery), totally collected retinal cells are supposedly to 105,500 cells. However, they only reported 56,531 total retinal cells collected. How that is possible?

RESPONSE: Thank the reviewer for the suggestion. We provide the numbers of individual retinal cell types before and after the injury (see Suppl. Table 1). When estimating the expected total number of retinal cells based on RGC counts, it is important to account for variations in the resilience of different retinal cell types to the dissociation and single-cell sequencing procedure. This variance arises from the structural fragility of certain cell types (e.g., photoreceptors), making them more susceptible to enzymatic and mechanical dissociation methods. Moreover, the removal of low-quality cells during computational analysis may disproportionately affect retinal cell types. Therefore, retinal cell types exhibit varying survival rates during the procedure, which can lead to inaccuracies when attempting to directly estimate the total cell count based solely on known population percentages of individual cell types. Due to the same reason, we are reluctant to over-interpret the number of individual retinal cell types before and after the injury.

2) Fig. 1b: Are there heterogeneity among muller glia? How about RodBC? Cones? HCs? The UMAP shows sub-structure in them, are they due to batch effect? Time points after injury? Quality issue? Or actual subtypes among types? The activated Muller glia shares some gene features with astrocyte. How are the two separated? Benhar et al. 2023 paper described in more details the gene changes from subtypes of muller glia/astrocyte/micro glia/ immune populations upon injury. It would be interesting to know if the subtypes were identified in the current data, and if the LR pairs showed any differences among them.

RESPONSE: Frist, we do observe the heterogeneity within certain clusters. For example, the cone bipolar cells (ConeBC) cluster can be subdivided into ON-ConeBC and OFF-ConeBC (see Figure below).

Moreover, the heterogeneity is not attributed to batch effects, as the UMAPs remain consistent across different time points (see Figure below).

In our data, Muller glia and astrocytes can be separated by their respective marker genes (see Figure 1d).

Second, it is important to emphasize that our intention is not to create a comprehensive retinal atlas by generating and characterizing the subtypes for each retinal cell types. Benhar et al. have already done a great job in identifying subtypes of the major retinal cell types and the temporal expression patterns of these subtypes. We do not want to replicate the effort in this work. Instead, the focus of our work is to identify cell-cell interactions between RGCs and other retinal cell types. We demonstrate the temporal patterns of LR pairs in Figure 2C and Fig. S2E. To identify the neuroprotective interactions, we have already generated ligand-receptor interaction data between 9 RGC subclasses and 13 retinal cell types, resulting in a substantial dataset comprising 117 pairs of cell types. If we were to further dissect the cell-cell interactions within each sub-cluster of every retinal cell type, the project would quickly become unmanageable. Lastly, we acknowledge that there may be important ligand-receptor interactions between subtypes of RGCs and subtypes of each retinal cell type. This is an avenue we plan to explore in our future projects, and any resulting findings will be subject to additional publications.

3) Fig. 1e: line 113-115 “Moreover, genes involved in protein translation and glycolytic metabolism were upregulated in almost all retinal cell types (Fig. 1e, Supplementary Fig. 1)”, It’s difficult to appreciate where are upregulated protein translation genes in the Fig. 1e and Suppl. Fig. 1, can you clarify? And it is well known RGC protein translation is shut off after optic nerve crush injury, better to show RGC data as well.

RESPONSE: We appreciate the reviewer for bringing up the concern. The sentence was intended to refer to Fig. 1f and Suppl. Fig. 1a-c, not Fig. 1e. We corrected the error. To improve the clarity, we have rephrased the sentence as follows: “Moreover, GO analyses have demonstrated a temporal increase in glycolytic activity, ribosomal subunit assembly, cytoplasmic translation, peptide biosynthesis, and oxidative phosphorylation processes within the two-day timeframe following ONC. This increase was observed in retinal ganglion cell (RGC) neighboring cell types, including Müller glial cells (MG), astrocytes, microglia, or amacrine cells (Fig. 1f and Suppl. Fig. 1a-c)... “

4) Fig. 1g: which groups are compared to generate the fold change, 12hr vs sham, 1d vs sham, or 2d vs sham? Or it is pooled together? What we can learn from the different time points? Are the ligand-receptor pairs are present in non-RGCs and RGCs separately to be considered as cell-cell interaction effect or did not differentiate at all?

RESPONSE: In this analysis, fold change values obtained from comparing 12 hours vs. sham, 1 day vs. sham, and 2 days vs. sham were combined. For each gene, the highest absolute $|\log_2FC|$ value among these three comparison groups was utilized in generating the figures. We made it clear in the Figure legend. Note that in this figure we did not investigate ligand-receptor interactions between RGCs and non-RGCs. Instead, we focused on the transcriptomic responses within non-RGCs and conducted a gene expression analysis separately for the ligand and receptor genes within these non-RGCs, without taking into account their interactions with RGCs. Our analysis of ligand-receptor interactions starts from Figure 2.

5) Suppl. Fig. 2a: the correlation is done with naïve RGCs or pooled naïve and injured RGCs? It needs more evidence showing the Tran et al 2019 dataset is a valid surrogate for this study. Not even sure this can be done this way. Suppl. Fig. 2a showed that many genes have different levels of expressions between the two datasets. More importantly, it's not clear if the injury induced changes are consistent, and therefore raise the question whether replacing the RGCs with those from another dataset is valid for the ligand-receptor analysis. More thorough comparison is needed. In principle, should be done within the same dataset from the same preparation.

RESPONSE: Please see our response in the first section. The correlation was done with all RGCs combined.

6) Fig. 2: overly simplified correlation analysis without considering potential non-RGC cell-cell signaling pairs and RGC autocrine effects.

RESPONSE: The focus of the paper is the interactions between the RGCs and other retinal cells. As suggested by the reviewer, we performed an analysis to identify the interactions between non-RGC cells. We incorporated the new results in Supplementary Fig. 10a, which included both autocrine and paracrine interactions. In addition, we also analyzed the RGC autocrine effects in Fig. 4e, showing that the protective interactions were achieved through both paracrine or autocrine signaling.

7) Fig2a. Does the number of cells in each class affect the ability to detect LR pairs? Any correlation between cell number and number of LR pairs?

RESPONSE: It is clear that the cell number does not affect the ability to detect LR pairs. For instance, astrocytes have a relatively small population of cells (Fig. 1b), yet we identified the highest number of L-R pairs with RGCs. Conversely, photoreceptors (rods and cones) have a larger cell number (Fig. 1b), but they establish the fewest L-R pairs with RGCs. Overall, there is no correlation between the cell numbers and LR pairs (see below). We included the new result in Supplementary Fig. 2b.

8) Fig2a. Which time point of the data was used to generate the figure? How do the number of LR pairs change along the time of injury?

RESPONSE: We calculated the total unique LR pairs across the four time points and showed the numbers in Fig 2a. The number of LR pairs remains relatively stable throughout the time course (see below). We included the result in Supplementary Fig. 2c.

9) Fig2a. It's surprising that many of LR pairs exist between photoreceptors and RGCs, or RPEs and RGCs. Are these real interactions or are these computational noises?

RESPONSE: The reviewer raised an interesting question. First, we want to highlight that the number of LR pairs detected between photoreceptors and RGCs is significantly lower than those between other cell types and RGCs. Even though our LRLoop algorithm does not incorporate spatial distribution information, the limited number of predicted LR pairs between photoreceptors and RGCs underscores the efficacy of the algorithms. Second, no prior studies have definitively stated the absence of interactions between these spatially separated cell types. Secreted ligands may still be able to migrate between these cell types. Last, our current computational methods cannot distinguish between direct and indirect interactions among cell types. It is possible that some interactions are mediated through intermediary cell types.

10) Fig2a/3a. Are the numbers normalized by the number of genes detected per cell?

RESPONSE: The number of L-R pairs and sum of S_{LR} are not normalized by the number of genes detected per cell.

11) Fig. 3d: "Validation of Anxa1-Dysf interaction, from MG to α RGC." Not sure how this figure validates interaction of Anxa2 and Dysf since these signals are from different cell types and layers and not even close to each other? Is Anxa2 secreted? It is a membrane protein, not secreted protein, right?

RESPONSE: First, it is well-established that Muller glia interact with neurons such as RGCs to regulate various biological processes (see a review "Glia-neuron interactions in the mammalian retina" PMID: 26113209). Although the cell bodies of Müller glia reside in the inner nuclear layer (INL), their endfeet extend into the ganglion cell layer (GCL), where the somas of RGCs are located.

Second, according to previous studies, Anxa1 can undergo externalized and/or secreted, functioning in an autocrine, paracrine and juxtacrine (involving cell-cell contact) manner to activate the downstream signaling (PMID: 19104500).

12) Fig3d bottom row, many Dysf+ cells are Spp1-, especially 2d post injury. Is this an α RGC-Muller interaction or a 2dpcRGC-Muller interaction or an otherRGC-Muller interaction?

RESPONSE: According to our scRNA-seq data, we identified Anxa1-Dysf as a ligand-receptor pair between Müller glia and α RGC. In Figure 3d, many non- α RGC cells also express Dysf. However, the expression of Dysf is most pronounced in α RGC, exhibiting significant changes following injury (see figure below). We made it clear in the revised manuscript.

13) Fig. 7b: normally microbeads model cannot sustain IOP elevation more than 3 weeks and often require 2nd injection, how this is achieved?

RESPONSE: We adopted a method from a previous published paper (see Figure 2 in PMID: 27077732), in which the IOP can sustain for at least six weeks. Briefly, we used magnetic microbeads for intracameral injection and used magnets to draw the microbeads towards the circumference between the cornea and iris. With this method, we successfully maintained elevated IOP for up to seven to eight weeks.

14) Fig. 7c: how about axon protection in optic nerve? If there is only RGC soma protection, it is hard to reconcile the visual function effect.

RESPONSE: We thank the reviewer for the insightful question. Our vision-guided behavior experiments have revealed that the overexpression of Oprm1 enhances vision in the glaucoma model, suggesting the potential protective effect on optic nerve. To validate the protective effect on optic nerve, we utilized toluidine blue staining to visualize myelinated nerve fibers. The figure illustrates cross-sections of the optic nerve, located 1 mm from the eyeball. The overexpression of Oprm1 resulted in a substantial improvement in axon survival in eIOP model, increasing from 66.5% to 84.3%. This finding aligns with the observed survival rate of retinal ganglion cell somas. We include the result in Fig. 7e and f.

15) Need to make statements more accurate.

For example, regarding line 70-71, “the optic nerve crush (ONC) model, which exhibits similarities in gene expression patterns with glaucoma17”, is certainly wrong. Crush model is an acute axonomy injury that is drastically different to ocular hypertension glaucoma and more so for normal tension glaucoma.

RESPONSE: Following the reviewer’s suggestion, we deleted the sentence.

In line 109, the result suggests that the downregulation of genes associated with chromatin assembly and remodeling is transit, however, Muller glia can’t spontaneously reprogramming after retinal injury in the mammalian, even when the expression levels of relevant genes bounce back 2d after injury.

RESPONSE: These are the genes enriched in our Gene Ontology analysis. We did not intend to imply that Müller glia spontaneously undergo reprogramming after retinal injury following the downregulation of these genes.

Throughout the manuscript, ligand-receptor interaction is used but without experimental proof that the interaction is really happening, it is just co-expression and potential pairs.

RESPONSE: Please see our response in the first section.

Reviewer #3 (Remarks to the Author):

This is a well-designed study which used a combination of bioinformatics tools, in vivo models of retinal injury and mouse behavioral assays to reveal a new role of Oprm1 in promoting the survival of RGCs.

The authors performed droplet-based scRNA-seq on the retina at three time points after the optic nerve crush (ONC) injury. The authors performed LRLoop, a recently developed computational method for analyzing ligand-receptor based cell-cell communication, to explore interactions among RGCs and the retinal cell types. Importantly, they identified the u-opioid receptor, encoded by the Oprm1 gene, as a novel neuroprotective factor for pan-RGCs. While the roles of u-opioid receptor in regulating pain, reward, and addictive behaviors are well characterized, its role in promoting RGC survival was not describe previously. The authors elegantly used AAVs and Cre mice to determine if Oprm1 is a neuroprotective factor for RGCs. Strikingly, expression of Oprm1 promoted survival of RGCs and prevented vision function loss in a clinically relevant model of glaucoma. The snRNAseq and the cell-cell interaction dataset provided in this study will be a valuable resource, not only for the investigators in the vision research field, but also for those who study neuroprotective strategies, neurodegeneration and neuronal death mechanisms in general. The data are presented with a high degree of clarity. There are a few issues which the authors could address to further strengthen the study.

RESPONSE: We thank the reviewer for the positive comments.

Tissue validation of Oprm1 expression (at the mRNA and protein levels) in ipRGCs is lacking. Studies have shown that most ipRGC subtypes (e.g. M1-M6 ipRGCs) have high survival capacity. Tran et al., scRNAseq dataset seem to show that M2 ipRGCs express high level of Oprm1. However, it is unclear if other ipRGC subtypes also express Oprm1. Perhaps the authors could use their own dataset or those from others to clarify the extent to Oprm1 expression in the different ipRGC subtypes.

RESPONSE: To address the concerns raised by the reviewer, we initially analyzed Oprm1 expression patterns across ipRGC subtypes (M1-M6) utilizing Tran's data. Our snRNA-seq using sorted RGC nuclei also indicated a preferential high expression of Oprm1 in ipRGCs M2 and M4 (also known as the alpha-ON-S) subtypes. Subsequently, our own snRNA-seq data validated these results (see figure below).

As M2 and M4 are primarily characterized by Opn4 and Spp1 marker genes, we conducted antibody staining in Oprm1 mCherry-tagged mice (See Figure below). The results revealed that ~80% of the Oprm1-expressing cells are either Opn4+ or Spp1+. This data is included as the new Fig. 6c and d, and Supplementary Fig. 10c-e.

It is unclear whether or not the AAV2-FLEX-sun1GFP and AAV2-FLEX-hOprm1 induce expression of transgene equally and proportionally in all RGC types. The authors need to use IHC (e.g. antibodies against RGC subtype markers) and characterize the extent of GFP expression in different RGC types.

RESPONSE: We agree with the reviewer that AAV2-FLEX-sun1GFP may not equally and proportionally induce transgene expression in all RGC types. We used it as a control virus, as existing studies did not suggest any neuroprotective effects associated with the overexpression of Sun1GFP.

As suggested by the reviewer, we performed the antibody staining with various RGC subclass marker. Our results shows that the AAV2-FLEX-hOprm1-mCherry induced high Oprm1 expression coverage of pan-RGC, and covering multiple RGC subclasses we examined (see Figure below). We include the result in Supplementary Fig. 8.

It is unclear how specific the Oprm1 ligands, proenkephalin (Penk), prodynorphin (Pdyn), and proopioidmelanocortin (Pomc) are in activating Oprm1 specifically. It is unclear if these ligands have other receptors which they can bind to. If there are other receptors which exist and if

they are indeed expressed in in retinal cells other than RGCs, then it is plausible that these ligands might act also on different retinal cells to induce RGC protection. In these regards, the mechanisms by which Oprm1 protects RGCs is nuclear. Perhaps the authors could elaborate and discuss these aspects.

RESPONSE: We have identified additional receptors such as Oprd1, Oprk1, Mrgprx1, and Mc1r, that interact with Penk, Pdyn, and Pomc. We cannot rule out the possibility that expression of these receptors in other non-RGC cells could be neuroprotective. Nevertheless, expression of these receptors in RGCs are unlikely to be neuroprotective based on our prediction, as they do not show significant differential interactions between high- and low-survival RGC subclasses. Note that in our study we directly manipulated Oprm1 in RGCs and showed that it had neuroprotective effects.

Given the recent studies which performed scRNAseq on human retinas, it will be useful if the authors can comment on whether or not Oprm1 is expressed in human RGCs and whether its ligands are expressed in RGCs or in non-RGC cells in the human retina. It remains a questions whether Oprm1 works in RGCs strictly or the similar mechanism operates in other CNS regions. While Oprm1 is expressed predominantly in ipRGCs in the retina, the pan-RGC protection in Oprm1 overexpressed animals indicate that the machinery downstream of Oprm1 is present in all neurons. The authors could elaborate and provide insights into expression level of Oprm1 and correlation of its natural expression to survival capacity of CNS neurons outside the retina.

RESPONSE: We have analyzed the gene expression profile of Oprm1 in the human retina [PMID: 32555229]. Interestingly, Oprm1 is predominately expressed in RGC5 cluster, which corresponds to ipRGC subclass. This finding suggests the potential applicability of the protective effect to human RGCs. Whether Oprm1 has protective effects on other CNS neurons would be an interesting topic to pursue in the future.

Several studies highly relevant to the topics discussed in this article could be cited and added to the reference list including, “Subtype-specific survival and regeneration of retinal ganglion cells in response to injury”. PMID: 36035999.

RESPONSE: We added the reference.

*Other minor issues are as follows;
Lines 140, 166, typos are found (i.e. ligand-recpetor).
Scale bars in some images are missing.*

RESPONSE: We fixed the typos and added the missing scale bars.